# HoK3v3: an Environment for Generalization in Heterogeneous Multi-agent Reinforcement Learning

Lin Liu♮, Jianzhun Shao†, Xinkai Chen‡, Yun Qu†, Boyuan Wang†, Zhenbin Ye♮, Yuexuan Tu♮, Hongyang Qin§, Lvfang Tao§, Junfeng Yang♮, Lin Lai♮, Yuanqin Wang♮, Meng Meng♮, Wenjun Wang♮, Lin Yuan♮, Xiyang Ji§, Minwen Deng§, Juchao Zhuo§, Qiang Fu§, Wei Yang§, Guang Yang♮, Lanxiao Huang♮, Wengang Zhou♭, Houqing Li♭, Ning Xie‡, Xiangyang Ji†, Deheng Ye§

♮Tencent Timi Studio, §Tencent AI Lab, ♭University of Science and Technology of China, †Tsinghua University, ‡University of Electronic Science and technology
{sjz18,qy22,wangby22}@mails.tsinghua.edu.cn, xyji@tsinghua.edu.cn,
202152011708@std.uestc.edu.cn, seanxiening@gmail.com,{lihq,zhwg}@ustc.edu.cn,
{lincliu,zhenbinye,tobytu,hongyangqin,luistao,fengjunyang,linlai,
yuanqinwang,promengmeng, jameswang,tayloryuan,xiyangji,danierdeng,
jojozhuo,leonfu,mikoyang,willyang,jackiehuang,dericye}@tencent.com

## Abstract

We introduce HoK3v3, a 3v3 game environment for multi-agent reinforcement learning (MARL) research, based on *Honor of Kings*, the world's most popular Multiplayer Online Battle Arena (MOBA) game at present. Due to the presence of diverse heroes and lineups (a.k.a., hero combinations), this environment poses a unique challenge for generalization in heterogeneous MARL. A detailed description of the tasks contained in HoK3v3, including observations, structured actions, and multi-head reward specifications, has been provided. We validate the environment by applying conventional MARL baseline algorithms. We examine the challenges of generalization through experiments involving the 3v3 MOBA full game task and its decomposed sub tasks, executed by lineups picked from the hero pool. The results demonstrate that HoK3v3 offers appropriate scenarios for evaluating the effectiveness of RL methods when dealing with the challenge of heterogeneous generalization. All of the code, tutorial, encrypted game engine, can be accessed at: https://github.com/tencent-ailab/hok_env.

## 1 Introduction

Multi-agent reinforcement learning (MARL) has shown great potential in many areas like robotics [11, 7], autonomous vehicles [21], industrial manufacturing [1], and social science [6]. However, deploying current MARL algorithms to real world scenarios still faces difficulty due to the large domain gap between simulators and real-world scenarios. This calls for: 1) stronger generalization ability for agents; and 2) more complicated and practical environments for MARL.

There are a series of existing standardized MARL environments supporting the research of this area. Multi-agent Particle Environment [10] is a simple yet effective simulator for particle-based agents moving on a 2D plane. The StarCraft Multi-Agent Challenge [14] gives a series of micromanagement challenges by wrapping the environment of real-time strategy game StarCraft II [18]. And Google Research Football [8] simulates physics-based football games with stochasticity. All these environ-

Submitted to the 37th Conference on Neural Information Processing Systems (NeurIPS 2023) Track on Datasets and Benchmarks. Do not distribute.

ments have small-sized action space or state space, and relatively simple tasks for the convenience of research. Therefore, these environments can hardly simulate the complicated real-world scenarios. In the meantime, agents usually share similar properties, only diverging from the numerical values, like the speed or hurt per frame of one agent. This makes the parameter sharing be an effective technology for current MARL algorithms, which actually trains some copies of one agent to handle multi tasks rather than training many agents to cooperate.

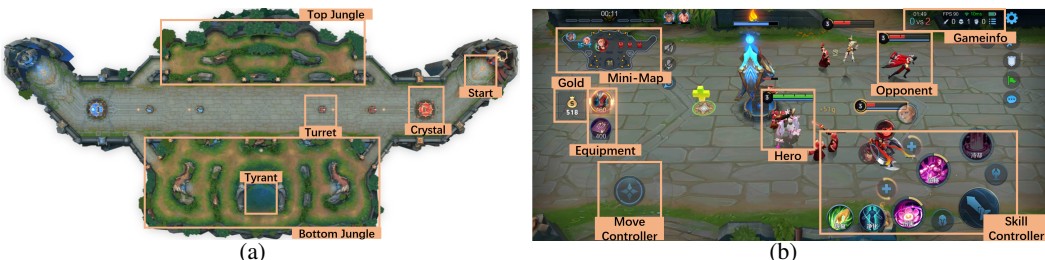

(a)                   (b)

Figure 1: (a) The map of `HoK3v3`, where we mark the locations of turrets, crystal and jungles. (b) Game user interface (UI). There are four sub-parts including a mini-map on the top-left, game information on the top-right, move controller on the bottom-left and skill controller on the bottom-right. Additionally, we use yellow boxes to highlight critical resources and units, namely gold and equipment, as well as heroes.

In this paper, we introduce the Honor of Kings 3v3 Arena (`HoK3v3`), a Multi-player Online Battle Arena (MOBA) environment that is authorized by the game Honor of Kings [1], with over 100 million daily active players [20]. There are two teams (or camps) (note that we use team or camp interchangeably) in the 3v3 environment, with each camp comprising a team of 3 heterogeneous agents, referred to as "heroes". To play a human-controlled game, each camp need 3 players, with each player controlling one hero via a smartphone. Agents within the same camp are expected to collaborate in order to secure victory by destroying the opponent's crystal. This 3v3 environment can be modeled as a mixed multi-agent task, wherein competition exists at the camp level, while cooperation is fostered within the camp.

For each agent, it can select one hero from the hero pool at the beginning of each game episode. Each agent assumes a distinct role within a camp, and different heroes possess varying action controls and agent attributes. These characteristics of the `HoK3v3` present the following challenges for MARL:

• A complicated multi-agent scenario that uses the same gamecore as the popular mobile game Honor of Kings, thus bearing resemblance to real-world scenarios.

• Generalization challenges within and across the team: 1) across the team: There exists 1000 different lineups, i.e., hero combinations. A well-trained multi-agent team policy must be capable of effectively handling all potential team lineups, while simultaneously adapting to any opponent's lineup. 2) within the team: Agents should learn to cooperate with teammates who have chosen different heroes.

Our contributions are summarized as follows:

• **Environment.** We propose Honor of Kings 3v3 Arena, a heterogeneous multi-agent environment with highly-optimized game engine that simulates the popular MOBA game, Honor of Kings.

• **API.** We provide standardized APIs for deploying MARL methods on `HoK3v3`. We abstract the complex game state with feature engineering and use several vectors of fixed length to represent the observations. A hierarchical structured action space is applied to cover all actions an agent can take.

• **Benchmark.** Apart from the full game (full task), we also give a series of easier tasks by decomposing the full game to evaluate the ability of the model trained with limited resources.

[1] https://en.wikipedia.org/wiki/Honor_of_Kings

Furthermore, we provide multiple pre-trained models with varying proficiency levels for the purpose of evaluation.

• **Baselines.** We evaluate some widely used MARL methods in HoK3v3 and give the results for future comparison.

• **Generalization.** We present an examination of the generalization challenges encountered in HoK3v3, showing it offers appropriate scenarios for evaluating the effectiveness of RL methods when dealing with the challenge of heterogeneous generalization

## 2 Characteristic and Related Work

The uniqueness of HoK3v3 is to provide a high-dimensional and heterogeneous multi-agent setting which has a hierarchical action space and a global task that can be explicitly factorized into several sub-tasks.

**Agent Heterogeneity.** There are many existing public environments for research on multi-agent reinforcement learning. Some focus on the cooperation between agents, like Google Research Football (GRF) [8], the StarCraft Multi-Agent Challenge (SC2) [14], and Multi-agent MuJoCo [12]. And some contains a series of multi-agent tasks including cooperation and competition like Multi-agent Particle Environment (MPE) [10] and Melting Pot [9]. However, they usually do not distinguish the function and action effect between agents, resulting in the homogeneity of trained agents. In contrast, the *hero* setting in HoK3v3 brings a large domain gap between different agents, requiring the robustness and generalization ability of the trained policy. The most related work to ours is HoK Arena [20], which is also built upon the HoK environment. However, this work primarily concentrates on competitive setting within a 1v1 MOBA game, without considering heterogeneity in cooperation. By comparison, we focus on heterogeneous teammates' cooperation in MARL and delve into a thorough investigation of the corresponding generalization problems. We summarize a detailed comparison of HoK3v3 and other related works in Appendix C.6.

**Structured Action Space.** The existing environments typically have a flat action space, which can be either discrete or continuous [10, 8]. However, it is also possible for the action space to exhibit a hierarchical structure. One commonly encountered type of structured action space is the parameterized action space [5, 4, 19, 3], where a discrete action is parameterized using a continuous real-valued vector. In our environment, we utilize a discretized parameterized action space, which we call hierarchical structured action space, to simplify the complex control involved.

**Factorizable Tasks.** Almost all environments for MARL provide a lot of tasks. The tasks in SC2 and GRF mainly differ in agents' property and quantity, while the target of each task in MPE and Melting Pot is designed to be different. The target of such tasks usually keeps integral and none of the current environments explicitly provide factorizable tasks, which lacks the support for hierarchical and goal-conditioned MARL. In addition, if the task is factorizable, it can better validate the performance of value decomposition [13, 16], which is an important research area for MARL. In contrast, in one game of HoK3v3, we have a global target to destroy the crystal of the enemy, which can be explicitly factorized into several sub-tasks, including gaining gold, killing enemies, destroying the defense turrets, etc. The global reward function is the summation of each sub-task's reward function, which makes the task decomposible and useful for MARL research mentioned above.

## 3 Honor of Kings 3v3 Arena Environment

The Honor of Kings 3v3 Arena is available as an open-source project under the Apache License V2.0, allowing individuals to engage in non-commercial activities. The code for agent training and evaluation has been developed with official authorization from Honor of Kings and can be found at: https://github.com/tencent-ailab/hok_env. Both the game engine and game replay tools are encrypted and comply with Tencent's Honor of Kings AI And Machine Learning

License 3[2]. Non-commercial users are welcome to register and freely download our game engine and tools. The documentary is available at: `https://doc.aiarena.tencent.com/paper/hok3v3/latest/hok3v3_env/honor-of-kings/`

## 3.1 Task Description

In the "Honor of Kings 3v3 Arena" game environment, players use the mobile button to control the movement of heroes, and use the skill button to control the heroes' normal attack and skills. The game environment has a fog of war mechanism, meaning that only the current units belonging to the friendly camp or within the observation range of the friendly camp can be observed. At the beginning of the game, the player controls the hero, starting from the base, to gain gold coins and experience by killing or destroying other game units (such as enemy heroes, creeps, and turrets), to buy equipment and upgrade skills, and thus enhance the ability of the hero. The winning goal is to destroy the opponent's turrets and base crystals, while protecting their own turrets and base crystals, Fig. 1.

## 3.2 Lineups

We use the term 'lineup' to represent the hero combinations in a game. In the game, there are 3 agents, each with a specific and unalterable role. Prior to the commencement of a game episode, each agent must select one hero from its relevant pool of candidates. These candidates vary in terms of their skills and properties. Each agent in a given camp has a distinct set of candidates with size as 10 heroes in this work, resulting in a camp with a total of $3 \times 10 = 30$ potential heroes. Consequently, each camp generates $10^3 = 1000$ possible lineups. When considering both camps together, the total number of lineups in a game amounts to $1000^2 = 10^6$, emphasizing the necessity for trained models to possess robustness and generalization capabilities. Fig. 2 illustrates an example of the lineups where each agent's candidate pool is limited to two or three heroes.

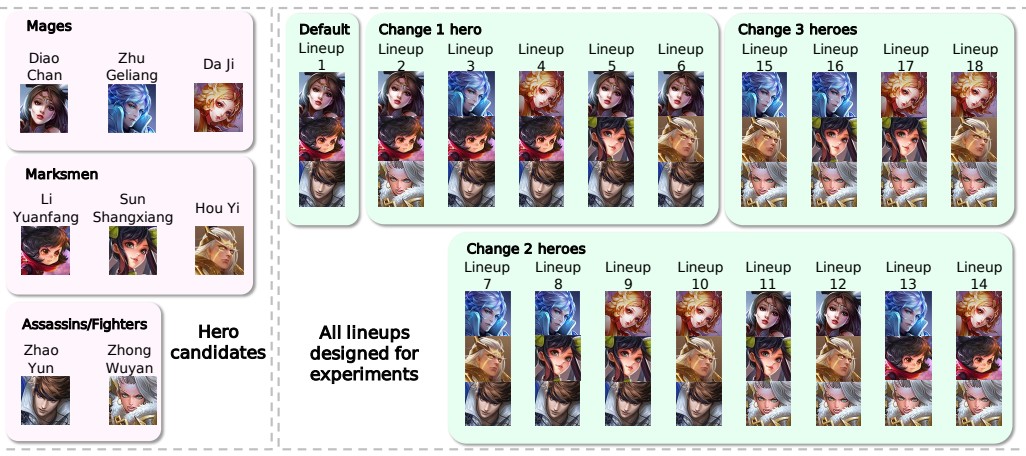

Figure 2: All hero candidates and lineups designed for experiments on generalization. For each role of agent (mages, marksmen and assassins/fighters), we choose from two or three heroes to build the lineup. The model is trained from one or two lineups and evaluated using all the lineups above.

## 3.3 Agents

Honor of Kings 3v3 Arena provides the same dimension of action space and state space for any hero. Here we give a simple description of each element of the MDP. See Appendix C for details.

**Observation Space** The observation space in `HoK3v3` is intricate, encompassing the essential status information within the game. Specifically, the observation space can be broken down into seven

---

[2]`https://github.com/tencent-ailab/hok_env/blob/master/GAMECORE.LICENSE`

primary components. *FeatureImgLikeMg* describes image-like features, including vision information. *VecFeatureHero* indicates the status information of the heroes. *MainHeroFeature* encompasses the current hero's private characteristics. *VecSoldier* describes the status of soldiers. *VecOrgan* provides information about the status of six turrets. *VecMonster* describes the status of monsters. *VecCampsWholeInfo* suggests situational characteristics. It is important to note that, except for *VecCampsWholeInfo*, all the features contain both absolute and relative information.

**Action Space** We employ a hierarchical structured action space to streamline the intricate control mechanisms which encompasses a hierarchical framework that covers all conceivable actions undertaken by the hero: 1) Which action button to choose; 2) How to operate specifically, such as controlling the direction of movement or skill drop point; 3) Which target to choose.

**Reward** The reward in HoK3v3 is calculated as a weighted sum of various configurations and is processed to be zero-sum by subtracting average reward of the enemy camp. Each of the configurations corresponds to one of the four distinct aspects: 1) Hero's farming related. 2) Kill-Death-Assist related. 3) Hero's own state related. 4) Game advancement related.

**Episode Dynamics** All heroes are initialized at their respective camp bases when an episode begins, and the termination condition of an episode in HoK3v3 is the destruction of any one of the crystals. In HoK3v3, actions are executed at a default interval of 133ms to match the response time of skilled amateur players. This interval can be modified as a configurable parameter. Moreover, during the training process, a predetermined time limit is imposed on episodes, while there are no time constraints in a regular round of the Honor of Kings game.

## 3.4 APIs and Implementation

For the facilitation of research demand, we encapsulate the original environment within a class named HoK3v3, which provides standardized APIs, as shown in Listing 1. The most crucial functions in this environment class are: *reset()* and *step()*. The former initiates a new episode, while the latter progresses the timeline based on a specified action. Both of them return a quadruple as follows:

- *obs_s*: A list of NumPy arrays containing observations of six heroes in two camps.

- *reward_s*: A list of floating-point numbers representing the processed immediate rewards associated with each hero.

- *done_s*: A list of two boolean values indicating the current termination state within the game.

- *state_dict_s*: A list of $Dict$ which contains additional information. The key elements within each $Dict$ include the following: *frame_no*, which represents the frame number of the next state; *player_id*, which identifies the runtime ID of the current hero; and two action masks, namely *legal_action* and *sub_action_mask*. For detailed information, please refer to Appendix C.

```python
from hok import HoK3v3

# load environment
env = HoK3v3.load_game(game_config)

# init agents
agents = [Agent(agent_config1), Agent(agent_config2)]

# rollout
obs_s, _, done_s, _ = env.reset()

while not any(done_s):
    action_s = []
    for i in range(env.num_agents):
        action = agents[i].process(obs_s[i])
        action_s.append(action)
    obs_s, reward_s, done_s, state_dict_s = env.step(action_s)
```

Listing 1: Python example

## 4  Validation

To assess the efficacy of HoK3v3, we conduct a series of experiments using a consistent lineup comprising *Zhaoyun, Diaochan, Liyuanfang* for both camps. In the subsequent sections, we present the baseline models employed, the evaluation metric utilized, and a comparative analysis of the performance exhibited by each model.

**Baselines** Due to the highly complex nature of the HoK3v3 and its structured action space, it is exceedingly challenging to directly apply conventional MARL algorithms commonly used in academia. Consequently, as a starting point, we utilize PPO [15] as our baseline algorithm, which has been validated as effective in similar environments [22, 20]. In order to tackle the issues of communication and reward decomposition in multi-agent learning, we also add two variants of PPO: a communication-based PPO, referred to as CPPO, and MAPPO [23]. In all methods, we employ a meticulously designed backbone as our feature extractor, specifically tailored to handle the extensive observation space. Additionally, each method is trained using the self-play technique to facilitate the discovery of novel and effective strategies. Please refer to Appendix G for details of the baselines. There is an integrated rule-based agent named *common-ai* within the Honor of Kings environment, which can be used to assess the effectiveness of baselines during the initial stages of training.

**Resources Requirement** To enhance the sampling process, policies are run in parallel over different CPU cores to generate samples. The standard training resource set consists of one NVIDIA Tesla T4 GPU with 600 CPU (Intel(R) Xeon(R) Platinum 8255C CPU @ 2.50GHz) cores for parallel training. However, it is also possible to utilize different computation resources in HoK3v3. In order to provide recommendations regarding resource requirements, keeping the GPU fixed, we conducted training of the CPPO network using varying numbers of CPU cores. The results are summarized in Table 1. It is evident that as the number of CPU cores increases, the training time experiences a significant decrease initially and gradually stabilizes thereafter with the increase of sample frequency and the decrease of consumption-generation ratio. Based on our experience, it is the consumption-generation ratio, i.e. the ratio of data consumption rate to generation rate, that ultimately determines performance. Therefore, we recommend researchers to maintain a consumption-generation ratio that is close to 1 when training CPPO in HoK3v3.

Table 1: Training results with varying computation resources, where **Training hours** represent the duration required to outperform *common-ai*, **Sample freq.** represents number of steps sampled per hour and **C-G ratio** denotes consumption-generation ratio.

| CPU Cores | Training hours | Sample freq. | C-G ratio |
|:---:|:---:|:---:|:---:|
| 64 | 54.30±2.84 | 5002.62±94.94 | 13.22±0.29 |
| 128 | 22.83±0.60 | 9495.90±187.04 | 6.92±0.12 |
| 256 | 5.62±0.80 | 20399.88±380.22 | 3.21±0.06 |
| 512 | 3.99±0.33 | 39812.00±527.74 | 1.63±0.03 |
| 600 | 3.67±0.30 | 46379.22±753.32 | 1.38±0.02 |

**Evaluation** We provide two kinds of evaluation metrics to measure the ability of a model.

• The winning rate against our pre-trained models. We provide six RL models with different levels (1-6) trained by CPPO. We reckon a model reaches level $i$ when it can achieve a winning rate larger than 50% against the preset level $i$'s model. Our approach for determining the different levels is based on the principle that level $i + 1$ should achieve a winning rate of at least 70% against level $i$. Table 2 presents the winning rates of each level when pitted against the other levels. The results demonstrate that a baseline model with a higher level exhibits an advantage over all lower levels, underscoring the robustness of our pre-trained models.

• The ELO score. It is often challenging for a team to break free from local optima if it only competes against a fixed opponent. To address this issue and assess the ability of models more precisely, we employ the Elo score [2]. The specific method for calculating the Elo score is explained

Table 2: The winning rate(%) of loop games between pre-trained models, and their Elo scores.

| Level | 1 | 2 | 3 | 4 | 5 | 6 | Elo Score |
|---|---|---|---|---|---|---|---|
| 1 | 50.0 | 26.6 | 10.9 | 0.0 | 0.0 | 0.0 | 971.47 |
| 2 | 73.4 | 50.0 | 16.4 | 0.8 | 0.0 | 0.0 | 1120.32 |
| 3 | 89.1 | 83.6 | 50.0 | 10.9 | 1.6 | 3.1 | 1408.34 |
| 4 | 100.0 | 99.2 | 89.1 | 50.0 | 22.7 | 15.7 | 1873.94 |
| 5 | 100.0 | 100.0 | 98.4 | 77.3 | 50.0 | 28.1 | 2110.65 |
| 6 | 100.0 | 100.0 | 96.9 | 84.3 | 71.9 | 50.0 | 2231.72 |

in Appendix D. As indicated in Table 2, the Elo scores accurately reflect the performance of the baselines. Figure 3(b) illustrates the Elo curves of the CPPO model during training, demonstrating an increasing trend and convergence after approximately 36 hours.

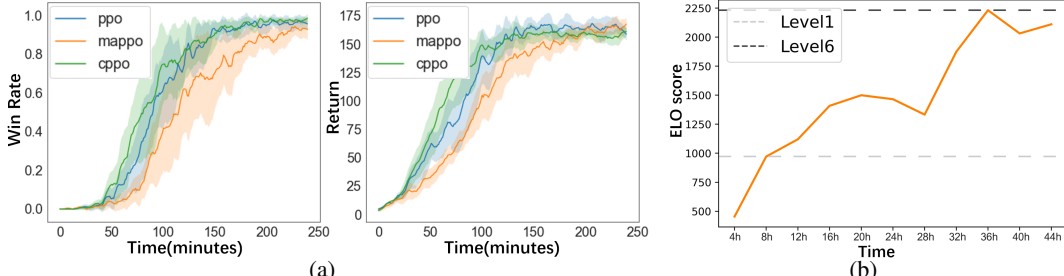

Figure 3: (a) The curves of *win_rate* and *return* with respect to the training time of baselines trained against *common-ai*. (b) The Elo curves with respect to the training time.

**Performance** We conduct experiments with 3 random seeds to evaluate the performance of baselines trained against the *common-ai* as shown in Figure 3(a). It can be observed that all the baselines outperform the *common-ai* within a short period, highlighting the efficacy of HoK3v3. While MAPPO falls behind the others, showing that independent learning[17] is better suited for the task.

## 5 Sub-tasks

The entire process in HoK3v3 can be naturally broken down into several sub-tasks. These sub-tasks encompass activities such as gaining golds, killing enemies and destroying turrets. Moreover, the overall reward is calculated as a weighted sum of the rewards associated with each sub-task, rendering the task decomposable.

Based on the nature of decomposability, we partition the HoK3v3 task into six sub-tasks, which are outlined below: **Gold**: collecting more golds. **Exp**: gaining more experience points. **Kill**: killing enemies as many times as possible. **Hurt**: inflict the highest possible rate of hurt. **Turret**: destroying the defense turrets. **Monster**: trying to attack monsters. Details of these sub-tasks will be introduced in Appendix F.

The results of the baselines for these sub-tasks are presented in Figure 4. Among the baselines, CPPO achieves the best performance, which can be attributed to the effective communication between agents that facilitates cooperation. Furthermore, compared to the original full game, the training time and computational requirements in sub-tasks are significantly reduced, enabling diverse research opportunities in our environment.

## 6 Generalization

In the HoK3v3, prior to commencing an episode, the agent possesses the chance to select various heroes to control, which constitutes a multitude of lineups. In such a scenario, training distinct models for each lineup individually would inevitably consume a large amount of time and prove impractical. Therefore, policies need to generalize on different heroes to adapt for various agent

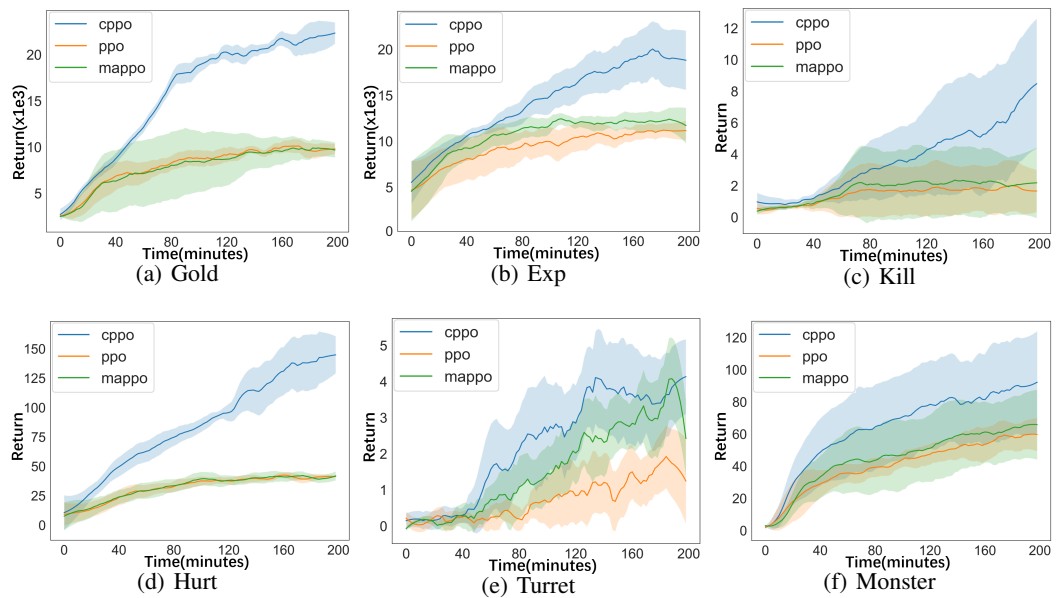

Figure 4: The results of the baselines on the sub-tasks. The maximum training time allowed is 200 minutes, and each experiment is conducted with three random seeds. It is evident that CPPO outperforms the other variants significantly across these sub-tasks.

and opponent lineups, thus leading the environment an advantageous platform for investigating the policies' generalization aptitude. In order to investigate generalization, we conduct experiments from two perspectives, (1) varying opponent lineups and (2) varying agent lineups. We build 18 lineups from the candidates shown in Fig. 2. Then we employ one fixed lineup, namely lineup 1 consisting of *{zhaoyun, diaochan, liyuanfang}*, to train the CPPO agent with self-play training until reaching level 6. Finally, the trained models are evaluated with different agent lineups or different opponent lineups, namely "Zero-Shot".

## 6.1 Varying Opponent Lineups

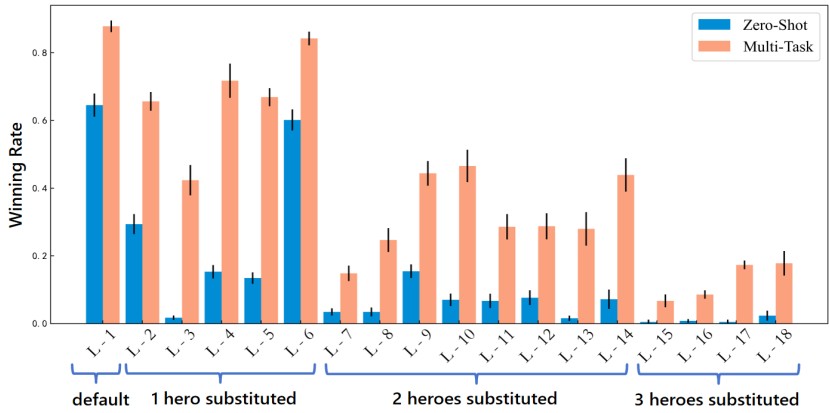

Figure 5: Generalization test on varying opponent lineups. *L-1* to *L-18* encompass a total of 18 lineups shown in Fig. 2. Blue: We train the model using a fixed lineup, namely *Lineup 1 VS Lineup 1*. Orange: We joint-train the model on *Lineup 1 vs Lineup 1 and 16*. We assess the performance of both models across 128 episodes in the scenario where *Lineup 1 VS Lineup 1-18* (Varying Opponent Lineups). The winning rate is evaluated across five random seeds.

We conduct several experiments to assess the generalization capabilities of models in the context of "Varying Opponent Lineups". As shown in Fig. 5, our findings reveal that the model trained on the

unaltered opponent lineup (*Lineup-1*) exhibit excellent performance, achieving a high win rate of 0.65. However, a significant drop in performance is observed when the opponent heroes are modified (*Lineup-2 to Lineup-18*). Moreover, the magnitude of performance degradation increase as more heroes are substituted, particularly when all three heroes are replaced (*Lineup-15 to Lineup-18*). These results indicate that existing methods face challenges in effectively addressing scenarios requiring generalization.

We aim to remedy this challenge by employing a "Multi-Task" approach, which replaces the opponents during training to *Lineup 1 and Lineup 16* thereby encompassing all the opponent heroes. As shown in Fig. 5, multi-task training improves the performance significantly in all the test tasks.

## 6.2 Varying Agent Lineups

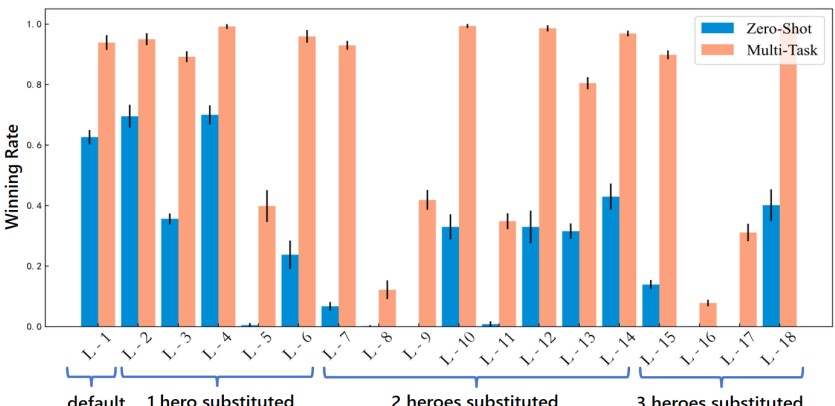

Figure 6: Generalization test on varying agent lineups. Details of lineups are similar to Fig. 5. Blue: We train the model using a fixed lineup, namely *Lineup 1 VS Lineup 1*. Orange: We joint-train the model on *Lineup 1 and 16 vs Lineup 1*. We assess the performance of both models across 128 episodes in the scenario where *Lineup 1-18 VS Lineup 1* (Varying Agent Lineups). The winning rate is evaluated across five random seeds.

Similarly to the "Varying Opponent Lineup" experiment, experiments are also carried out in the context of the "Varying Agent Lineup" to assess the capability of generalization of models in controlling different lineups while battling against the same opponent *Lineup-1*. The findings, as shown in Fig. 6, indicate that in certain instances (*Lineup-2 and Lineup-4*), models exhibit commendable generalization capabilities. However, in the majority of cases, models display limited generalization abilities when it comes to controlling diverse heroes. Furthermore, a significant drop in performance is observed when attempting to generalize to *Marksmen* heroes, underscoring the necessity for further research into algorithms that can enhance their generalization capabilities.

We also try "Multi-Task" approach to remedy this challenge. While training, we train the model with 2 tasks *Lineup-1 and Lineup-16 VS Lineup-1* thereby encompassing all the heroes to be controlled. As shown in Fig. 6, multi-task training improves the performance significantly in all the test tasks.

## 7 Conclusion

In this paper, we propose HoK3v3, a new environment for MARL research. We provide a comprehensive description of the environment and explain its implementation and APIs. By conducting a series of experiments using baseline algorithms, we validate its efficacy. Additionally, we decompose the full game into several easier sub-tasks to cater to diverse demands and limited computation resources. Furthermore, the presence of heterogeneous heroes and distinct roles in the environment provides scenarios and requirements for generalization. This environment is openly accessible for research purposes, and we anticipate diverse research initiatives based on HoK3v3.

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
