# A  Author Statement

We have released `HoK3v3` - the Honor of Kings 3v3 Arena - as an open-source project under the Apache License V2.0. The relevant code and gamecore can be found at `https://github.com/tencent-ailab/hok_env`, while detailed documentation for `HoK3v3` is available at `https://doc.aiarena.tencent.com/paper/hok3v3/latest/hok3v3_env/honor-of-kings/`. All experiments can be replicated using the source code, inclusive of hyperparameters and configurations. The game developers of Honor of Kings have authorized `HoK3v3`, and the authors bear all responsibility in case of violation of rights. We will also ensure data accessibility and provide necessary maintenance.

# B  Environment Details

## B.1  Mechanisms

There are two types of mechanisms present in `HoK3v3`: *Crystal* and *Turret*. Moreover, the Arena contains two *Turrets* and one *Crystal*. The primary objective for players is to engage in combat to safeguard their own *Crystal* while simultaneously attempting to destroy the opposing team's *Crystal*. The *Crystal* possesses a 90% resistance to damage, and it appears in the map within 10 seconds of the game's initiation.

*Turrets* are classified into two types: the *Vanguard Turret* and the *Base Turret*. Once the *Vanguard Turret* is destroyed, players can proceed to destroy the *Base Turret*, followed by the final *Crystal*. Both the *Vanguard Turret* and the *Base Turret* are formed 10 seconds after the game begins. For the initial 2 minutes, a protection mechanism is in place to safeguard the *Turrets*, allowing them to withstand 80 points of normal attack damage from heroes. Additionally, *Turrets* enjoy a 55% damage-free rate in the absence of *Creeps*.

Sixty seconds after the commencement of the game, an HP pack will become available behind both the *Vanguard Turret* and the *Base Turret*, providing heroes with the means to restore their health points (HP). Once utilized, the HP pack will disappear and subsequently replenish itself every 60 seconds. However, in the event that either *Turret* is destroyed, the regeneration of the HP pack of this *Turret* itself will cease.

The defense attributes of the mechanisms are shown in Table. 3.

Table 3:  Defense attributes of the mechanisms.

| Mechanisms | Basic HP | Growth HP | Basic Armor | Basic Resistance |
|---|---|---|---|---|
| *Vanguard Turret* | 6000 | 700 | 200 | 200 |
| *Base Turret* | 10000 | 700 | 200 | 200 |
| *Crystal* | 8000 | 600 | 200 | 200 |

The mechanisms select attack targets according to a consistent rule. If an enemy hero fails to inflict damage on ally heroes within the *Turret*'s attack range, the *Turret* will prioritize attacking the first unit that enters its range. Once the initial unit is eliminated, the *Turret* will then proceed to attack minions, summoned creatures, and heroes, in that order. In cases where units share the same priority, the *Turret* will direct its attacks towards the nearest unit. However, when an enemy hero inflicts damage on ally heroes, the *Turret* will focus on the first enemy hero responsible for the damage. This targeting persists until the enemy hero either exits the attack range or is eliminated.

The extent of damage inflicted by the mechanisms will accumulate with each subsequent attack, and this damage is characterized as physical damage that bypasses any defensive measures. The attacks performed by these mechanisms are listed in Table. 4.

Destroying enemy mechanisms will gain experience and golds, as shown in Table. 4.

Table 4: Attacks performed by mechanisms.

| Mechanisms | Basic Attack | Attack Bonus | Maximum Attack Bonus | Experience | Golds |
|---|---|---|---|---|---|
| *Vanguard Turret* | 430 | 300 | 1500 | 100 | 100 |
| *Base Turret* | 500 | 300 | 1500 | 100 | 100 |
| *Crystal* | 580 | 300 | 3000 | 0 | 0 |

## B.2 Creep

The creep serves as the primary source of experience and gold in the `HoK3v3`, constituting the largest proportion of these resources within the entire game. It is primarily categorized into two types: ordinary creep and super creep. Upon the player's destruction of the *Base Turret*, the ordinary creep is substituted with the more formidable super creep. Creep materializes 12 seconds after the commencement of the game and subsequently regenerate every 24 seconds.

In the first four minutes of the game, the composition of the creep consists of two *Warriors* and two *Mages* respectively. As the game progresses, the creep composition changes to include two *Warriors*, one *Mage*, and one *Catapult*. Upon the player's destruction of the *Base Turret*, the *Ordinary Creep* is replaced by the more formidable *Super Creep*, which is comprised of two *Warriors*, one *Mage*, and one *Super Warrior*. The essential attributes of the creep are presented in Table 5.

Table 5: Basic attributes of creep.

| Creep | Attack | Magic | Armor | Resistance | HP | Experience | Golds |
|---|---|---|---|---|---|---|---|
| *Warrior* | 60 | 60 | 0 | 0 | 1860 | 60 | 48 |
| *Mage* | 120 | 120 | 0 | 0 | 1545 | 45 | 36 |
| *Catapult* | 192 | 192 | 0 | 0 | 2790 | 100 | 84 |
| *Super Warrior* | 576 | 576 | 183 | 183 | 4185 | 100 | 70 |

## B.3 Jungles

The jungles serve as the primary source of experience and gold for *Assassin Heroes* such as "*Zhaoyun*". In this particular map, both sides' players have access to the entire jungle area. The jungles contain various types of monsters, including normal creatures, the formidable "*Scarlet Statue*," the elusive "*Treasure Thief*", and the powerful "*Tyrant*". The spatial distribution of various monster species within the jungles can be referenced in Figure. 7. Upon slaying the "Dark Wolf," a hero receives the "Forest's Roar" buff, granting them a 30% increase in movement speed and a 20% reduction in skill cooldowns for a duration of 30 seconds. However, this buff dissipates upon the hero's death.

Defeating the "*Tyrant*" bestows the hero with the "*Tyrant*'s Power." Accumulating multiple "*Tyrant*" kills further enhances this power. The effects of slaying the "*Tyrant*" multiple times are as follows: the first kill increases the HP recovery of all allies by 1% every 2 seconds, the second kill augments the damage inflicted by all allies against enemy mechanisms, and the third kill amplifies the physical and magical output of all allies by 30%. For additional information regarding the specifics of the jungles, please refer to Table 6.

Table 6: Details about the jungles.

| Creature Name | Attack | Magic | Armor | Resistance | HP | Experience | Golds |
|---|---|---|---|---|---|---|---|
| *Big / Little Demon Vanguard* | 138 82 | 138 82 | 183 109 | 183 109 | 2480 1488 | 60 30 | 60 30 |
| *Big / Little Archer* | 138 82 | 138 82 | 183 109 | 183 109 | 2480 1488 | 60 30 | 60 30 |
| *Big / Little White-tail Deer* | 138 82 | 138 82 | 183 109 | 183 109 | 2480 1488 | 60 30 | 60 30 |
| *Big / Little Dark Wolf* | 138 82 | 138 82 | 183 109 | 183 109 | 2480 1488 | 70 50 | 70 50 |
| *Scarlet Statue* | 216 | 216 | 183 | 183 | 3720 | 90 | 90 |
| *Treasure Thief* | 204 | 204 | 183 | 183 | 5400 | 160 | 135 |
| *Tyrant* | 204 | 204 | 183 | 183 | 9000 | 300 | 200 |

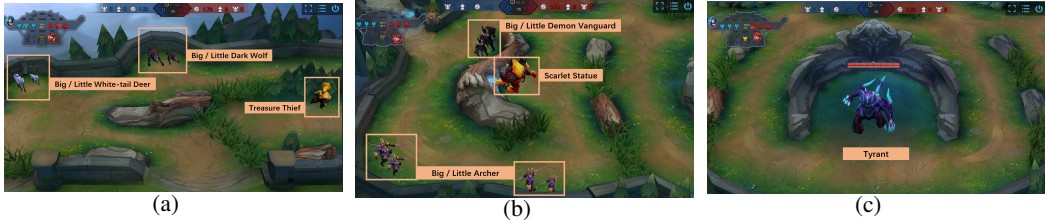

Figure 7: The spatial distribution of various monster species within the jungles.

## B.4 Heroes

In the HoK3v3, we have open-sourced a total of 30 heroes (Fig. 8), which can be classified into three types: *Mage*, *Marksmen*, and *Assassin*. Each type consists of 10 heroes. *Mage* and *Marksmen* heroes primarily operate in the middle lane, acquiring experience and gold by eliminating opponent heroes or creeps. On the other hand, *Assassin* heroes predominantly operate in the jungle, killing monsters to obtain gold and experience. Additionally, *Assassin* heroes also venture into the middle lane to collaborate with *Mage* and *Marksmen* heroes in eliminating opponent heroes or destroying *Turrets*. Each agent is able to select one hero to control and cooperate with the other two agents. For further details regarding hero skills, please consult the website: *https://pvp.qq.com/m/m201706/heroList.shtml*.

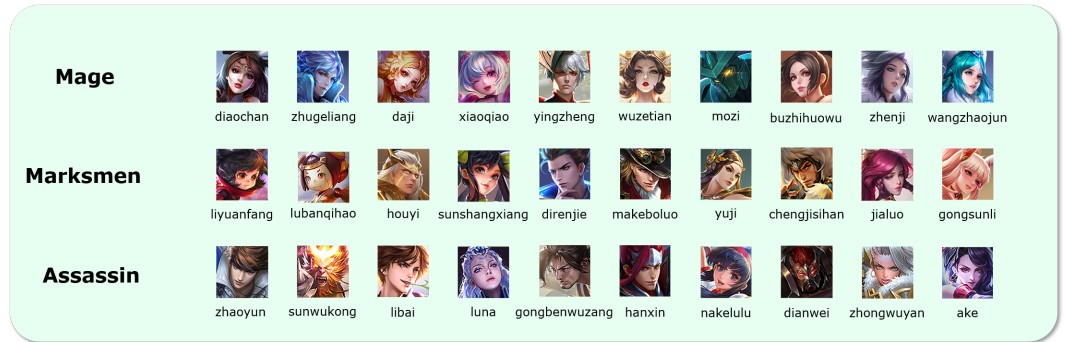

Figure 8: Details of 30 open-sourced heroes.

## C  Agent Details

### C.1  Observation Space

In order to enhance the training process and facilitate academic exploration, the HoK3v3 incorporates various components, including the intricate game interaction logic, the training framework project, and the feature design project. By encapsulating these elements, the HoK3v3 offers comprehensive and essential information. The complete observation space dimension comprises 4586, and Table 7 provides a detailed breakdown of feature categories, descriptions, and dimensions.

**Note:**

• All the features, except *FeatureWholeInfo*, consist of both absolute information features and relative information features. Let us consider *FeatureHero* as an illustrative example. The absolute information features, such as hero ID, blood, HP, and attack power, remain consistent for all three heroes. On the other hand, the relative information features, such as the x-axis, z-axis, distance, and other dimensions pertaining to the current player, vary among the three heroes.

• The units related to the two sides are presented in the following order: our side and enemy side. For example, the *FeatureHero* unit represents [3 heroes on our side, 3 heroes on the enemy side],

the *FeatureSoldier* unit represents [10 soldiers on our side, 10 soldiers on the enemy side], and the *FeatureOrgan* unit represents [3 turrets on our side, 3 turrets on the enemy side].

- Units that are not associated with specific sides are arranged according to the ID of the unit. For example, the *FeatureMonster* unit consists of [*Monster* 0, *Monster* 1, ..., *Monster* $N$], with the *Tyrant* unit being positioned last within the *Monster* unit.

Table 7: The categories of features, descriptions, and the dimensions.

| Categories of Features | Descriptions | Dimensions | Start Index | End Index |
|---|---|---|---|---|
| *FeatureImg* | Image-like features, including 6 channels such as obstacle channel and grass channel. | 6*17*17 | 0 | 1733 |
| *FeatureHero* | From vision of the current player, state information of 6 heroes from both sides, i.e. hero ID, HP, etc. | 6*251 | 1734 | 3239 |
| *FeatureMainHero* | Private features of the current hero, i.e. whether the enemy hero is within the attack range of the current hero. | 44 | 3240 | 3283 |
| *FeatureSoldier* | The state of 20 *Creep* of allies and enemies: types, HP, positions, etc. | 20*25 | 3284 | 3783 |
| *FeatureOrgan* | The state of 6 *Turrets*: types, HP, positions, etc. | 6*29 | 3784 | 3957 |
| *FeatureMonster* | The state of 20 *Monsters*: types, HP, positions, etc. | 20*28 | 3958 | 4517 |
| *FeatureWholeInfo* | Golds of allies and enemies; kills, surviving turrets, etc. | 68 | 4518 | 4585 |

## C.2  Action Space

The original action space in the HoK3v3 comprises a triad of action buttons: the move direction button, the skill offsets on the x- and z-axes button, and the target game units button. This comprehensive set encompasses all possible actions that the hero can undertake in a hierarchical fashion.

Specifically, the player must make decisions regarding the following aspects:

**Selection of action button**: The player needs to determine which action button to choose, such as the move button, attack button, skill button, return button, and so on.

**Execution details**: The player must specify the precise execution details, including controlling the direction of movement and managing the landing position of skills.

**Target selection**: The player must decide which target to select for the intended action.

Details of the action space can be referred to Table. 8.

## C.3  Legal Action Mask

As shown in Table 8, in each time step of an episode, every hero has the option to choose one action. However, their choice of action is not arbitrary. Therefore, a legal action mask exists for each time step, restricting the hero from selecting illegal actions. The dimension of the legal action mask is the same as that of the action when it comes to the action types "*Button*", "*Move*", and "*Skill*". However, there is a dependency relationship in the legal action mask between the action types "*Target*" and "*Button*": the legal actions of "*Target*" depend on the chosen "*Button*". In other words, only when a hero chooses a "*Button*" can they determine the legal actions for "*Target*". Since there are a total of 13 "*Buttons*" and 7 "*Targets*", the dimension of the legal action mask for "*Target*" is $13 \times 7$.

Table 8: Details of the action space.

| Action Type | Sub Action | Description | Dimension |
|---|---|---|---|
| *Button* | None | inactive state | 1 |
| | None | inactive state | 1 |
| | Move | move hero | 1 |
| | Normal Attack | cast normal attack | 1 |
| | Skill 1 | cast skill 1 | 1 |
| | Skill 2 | cast skill 2 | 1 |
| | Skill 3 | cast skill 3 | 1 |
| | Skill 4 | cast skill 4 (for specific heroes) | 1 |
| | Chosen Skill | cast chosen skill | 1 |
| | Recall | return to the base | 1 |
| | Equipment Skill | cast equipment skill | 1 |
| | Heal Skill | cast heal skill | 1 |
| | Friend Skill | cast friend skill (for specific heroes) | 1 |
| *Move* | Move Dir | move direction | 25 |
| **Skill** | Skill X | skill offsets on the x-axis | 42 |
| | Skill Z | skill offsets on the z-axis | 42 |
| *Target* | None | no target | 1 |
| | Enemy | 3 enemy heroes | 3 |
| | Friend | 3 friend heroes | 3 |
| | Self | own hero | 1 |
| | Monster | 20 monsters | 20 |
| | Soldier | 10 closest monsters | 10 |
| | Turret | the closest turret | 1 |

### C.4 Sub Action Mask

Action masking refers to the process of removing certain actions that cannot be executed simultaneously with the current action, resulting in a selection of permissible actions. To facilitate comprehension, we present a couple of examples:

**Example-1:** Upon choosing the Button-Move action, only the Move Dir sub-actions remain after the mask is applied. These sub-actions enable control over the direction of movement, as depicted in Fig. 9.

**Example-2:** Upon selecting the Button-Normal Attack action, only the Target sub-action remains, defining the target of the normal attack after the mask is applied.

Similar masking principles apply to other actions as well. It is important to note that the specific sub-action masks may vary depending on the heroes and equipment involved. For further information, please consult the official website at "*https://pvp.qq.com/web201605/herolist.shtml*".

### C.5 Reward

The design of rewards takes the following factors into consideration:

- Hero Development: The rewards are based on the golds and experience gained through killing monsters and creeps.

- KDA (Kills, Deaths, and Assists): The rewards are influenced by the player's performance in terms of kills, deaths, and assists.

- Hero's State: The rewards are tied to the hero's remaining HP (Health Points).

- Game Progression: The rewards are determined by the destruction of Turrets and the Crystal.

For specific details regarding the design of rewards, please refer to Table 10.

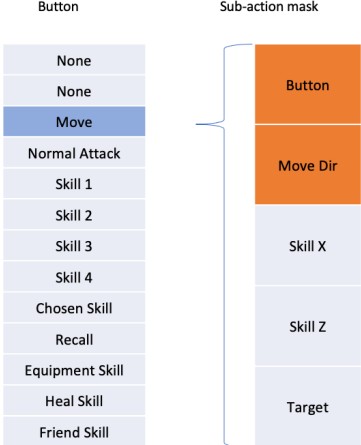

| Reward Factors | Descriptions | Weights |
|---|---|---|
| *hp_rate_sqrt_sqrt* | the fourth root of Hp rate | 1.0 |
| *money* | economic growth | 0.001 |
| *exp* | experience growth | 0.001 |
| *tower* | HP of own *Turret* | 1.0 |
| *killCnt* | kill count | 1.0 |
| *DeadCnt* | dead count | -1.0 |
| *assistCnt* | assist count | 1.0 |
| *total_hurt_to_hero* | damage to heroes | 0.1 |
| *atk_monster* | attack monster | 0.1 |
| *win_crystal* | destroy enemy crystal | 1.0 |
| *atk_crystal* | attack enemy crystal | 1.0 |

Figure 10: The specific design of rewards.

Figure 9: Sub Action mask after selecting the Button-Move action.

When it comes to calculating the final reward, we employ a "zero-sum" approach wherein the average reward of the enemy camp is subtracted. The individual hero reward is determined by applying weights, which can be referenced in Table 10. As demonstrated in Equation 1, the final reward is obtained by utilizing each hero's $hero_{reward_{zero\_sum}}$ value.

$$hero_{reward} = w_1 r_1 + w_2 r_2 + ... + w_n r_n$$

$$team_{reward} = \frac{1}{3} \sum_{i=1}^{3} hero_{reward}$$

$$hero_{reward_{zero\_sum}} = \begin{cases} hero_{reward} - team_{reward_{camp2}}, & \text{if hero in camp1} \\ hero_{reward} - team_{reward_{camp1}}, & \text{if hero in camp2} \end{cases}$$

(1)

### C.6 Comparison with related works

Table 9: A detailed comparison of `HoK3v3` and other related works.

| | Observation Space | Action Space | Focus | Heterogeneity[†] |
|---|---|---|---|---|
| **Google Research Football** | 115[*] | 19 | Cooperation | ✗ |
| **StarCraft Multi-Agent Challenge** | 16-336 | 7-70 | Cooperation | o |
| **Multi-agent MuJoCo** | ≤376 | ≤17 | Cooperation | o |
| **Multi-agent Particle Environment** | ≤20[**] | ≤10[**] | Cooperation&Competition | ✗ |
| **Melting Pot** | 88x88x3 | 6[***] | Cooperation&Competition | ✗ |
| **HoK Arena** | 491 | 83 | Competition | ✓ |
| **HoK3v3 (Ours)** | 4586 | 161 | Cooperation&Competition | ✓ |

[†] ✗ symbolizes homogeneity, "o" indicates only numerical heterogeneity, and ✓ denotes true heterogeneity.
[*] The floats representation proposed in the original paper.
[**] Estimations for common scenarios.
[***] Common movement actions proposed in original paper.

## D Elo Details

The Elo rating system is a common ranking system used in competitive matches[2]. Here is the calculation method of the Elo rating system used in this article:

1. Each model has an Elo rating, which represents their skill level in the competition. The initial Elo rating of each model is 1500.

2. In each match, the expected probability of winning for each model is calculated based on their Elo rating difference. The formula for expected probability is:

$$E_a = \frac{1}{1 + 10^{(\frac{R_b - R_a}{400})}} \tag{2}$$

$$E_b = 1 - E_a \tag{3}$$

where '$R_a$' and '$R_b$' are the ELO ratings of the two models.

3. After each match, the Elo ratings of the two models are updated based on the actual result. If model A wins, its actual score $S_a$ is 1 and model B's actual score is 0. The formula for updating Elo rating is:

$$R_a^{'} = E_a + K \times (S_a - E_a) \tag{4}$$

$$R_b^{'} = R_b + K \times ((1 - S_a) - E_b) \tag{5}$$

where K is a constant that determines the amount of change in Elo rating after each match. In this article, K = 40.

4. In this article, each model plays 128 matches against other models, and the win-loss records are shuffled randomly. The Elo ratings are updated based on the shuffled win-loss records, and this step is repeated for 200 times to calculate the average Elo ratings. This is done to reduce the error caused by different match orders.

# E Hyperparameters

We have included Table. 10, 11, 12 and 13, which present the key hyperparameters utilized in the relevant experiments. This table encompasses the essential parameters required for conducting the experiments effectively.

# F Sub-tasks Details

In these sub-tasks, we modify the reward function to individual reward item that corresponds specifically to the given sub-task, with weight as 1, in stead of weighted sum of multiple items. The concrete details of sub-tasks are as follows:

- **Gold**: Obtaining more gold generally provides a significant advantage, as it is the most important resource in the game. The objective of this sub-task is to collect more gold by destroying enemy units (heroes, creeps, and turrets) or monsters. The corresponding reward item for this sub-task is 'money'.

- **Exp**: Similar to gold, experience points are crucial in the game as they determine the level of the heroes. Therefore, we have designed this sub-task to modify the objective to focus on gaining more experience points, which helps heroes level up faster. The corresponding reward item for this sub-task is 'exp'.

- **Kill**: As a competitive game, killing an enemy provides both gold and experience points, while also temporarily incapacitating the slain enemy, thus granting a significant advantage to the team. Consequently, we have designed the **Kill** sub-task to specifically train the agents to eliminate enemies as frequently as possible. The corresponding reward item for this sub-task is 'killCnt'.

- **Hurt**: The rate of hurting is another metric that signifies killing enemies, and it carries a more dense reward. This sub-task serves as an alternative to the **Kill** objective, aiming to maximize the extent of hurt inflicted. The corresponding reward item for this sub-task is 'total_hurt_to_hero'.

- **Turret**: Destroying the turrets of enemies is a crucial sub-goal in the game, as it grants access to the crystal. We have designed the **Turret** sub-task to enhance players' abilities in destroying enemy turrets and defending themselves. The corresponding reward item for this sub-task is 'tower'.

Table 10: Hyperparameters.

| Hyperparameters | Value |
|---|---|
| Batch Size | 288 |
| $\gamma$ | 0.995 |
| LSTM Time Steps | 16 |
| $\lambda$ in GAE | 0.95 |
| PPO Clip $\epsilon$ | 0.2 |
| PPO Clip $c$ | 3.0 |
| Optimizer | Adam |
| beta1 | 0.9 |
| beta2 | 0.999 |
| eps | 1.00E-08 |
| Learning Rate | 6.00E-04 |

Table 11: Reward weight of "*Mages*".

| Reward Factors | Weights |
|---|---|
| *hp_rate_sqrt_sqrt* | 3.0 |
| *money* | 0.005 |
| *exp* | 0.0 |
| *tower* | 1.0 |
| *killCnt* | 1.0 |
| *DeadCnt* | 0.0 |
| *assistCnt* | 1.0 |
| *total_hurt_to_hero* | 0.3 |
| *atk_monster* | 0.0 |
| *win_crystal* | 0.0 |
| *atk_crystal* | 0.0 |

Table 12: Reward weight of "*Marksmen*".

| Reward Factors | Weights |
|---|---|
| *hp_rate_sqrt_sqrt* | 3.0 |
| *money* | 0.005 |
| *exp* | 0.0 |
| *tower* | 1.0 |
| *killCnt* | 1.0 |
| *DeadCnt* | 0.0 |
| *assistCnt* | 1.0 |
| *total_hurt_to_hero* | 0.3 |
| *atk_monster* | 0.0 |
| *win_crystal* | 0.0 |
| *atk_crystal* | 0.0 |

Table 13: Reward weight of "*Assassins*".

| Reward Factors | Weights |
|---|---|
| *hp_rate_sqrt_sqrt* | 3.0 |
| *money* | 0.005 |
| *exp* | 0.0 |
| *tower* | 1.0 |
| *killCnt* | 1.0 |
| *DeadCnt* | 0.0 |
| *assistCnt* | 1.0 |
| *total_hurt_to_hero* | 0.3 |
| *atk_monster* | 0.02 |
| *win_crystal* | 0.0 |
| *atk_crystal* | 0.0 |

• **Monster**: Monsters residing in the jungle play a vital role in enabling heroes to acquire gold and experience points. Instead of the original objective, we have modified it to focus on attacking monsters. The corresponding reward item for this sub-task is 'atk_monster'.

# G Baseline Details

**Encoder**: We employ a meticulously designed backbone as our feature extractor, specifically tailored to handle the extensive observation space. The encoder consists of multiple units dedicated to processing different aspects of the observation. These units encompass a convolution module, responsible for extracting image-like features, as well as distinct modules for hero, creep, turret, monster, and game status information. Please refer to the code `https://github.com/tencent-ailab/hok_env` for the implementation details of the network.

**PPO**: We employ Proximal Policy Optimization (PPO) [15] as our baseline algorithm. Specifically, we adopt the dual-clip version of PPO, which has been empirically validated as effective in similar environments [22, 20]. Additionally, to address the challenges posed by multi-agent control, we incorporate the independent learning paradigm [17] and parameter sharing with PPO.

**CPPO**: CPPO is a communication-based variant of PPO. This approach shares similarities with the standard implementation of PPO, with the exception that a portion of the processed feature from each hero undergoes max-pooling to obtain a shared feature. This shared feature is then utilized by both the policy and value networks.

**MAPPO**: To evaluate the efficacy of the CTDE paradigm, we employ a CTDE variant of PPO, known as MAPPO [23]. In contrast to independent learning, MAPPO incorporates a unified global value

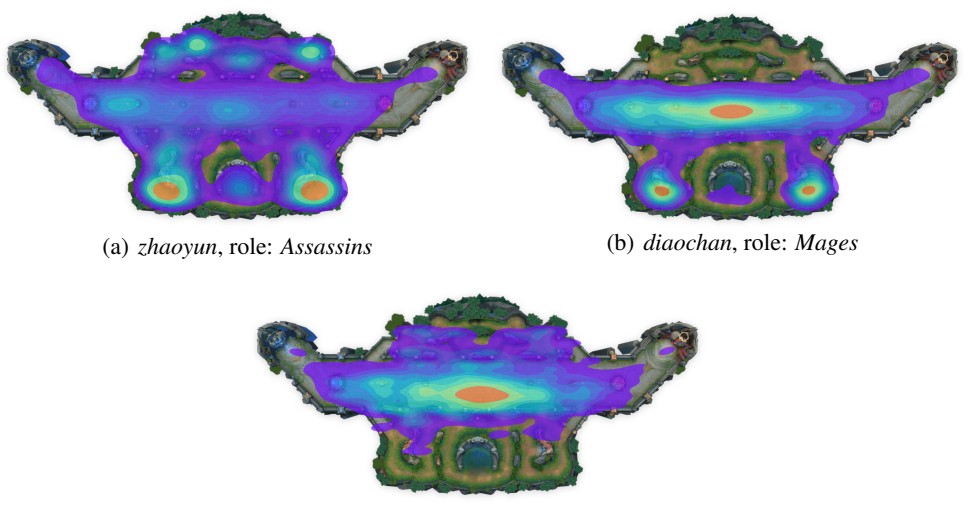

(a) *zhaoyun*, role: *Assassins*

(b) *diaochan*, role: *Mages*

(c) *liyuanfang*, role: *Marksmen*

Figure 11: The trajectories of heroes with different roles.

network that can access the processed features of all heroes and utilizes the average reward of the heroes within a team as a shared reward.

## H    Additional Experiments

### H.1    Visualizing Hero Trajectories

For the purpose of better illustrating the trajectories of heroes with different roles, namely *Assassins*, *Mages*, and *Marksmen*, we have constructed a heatmap utilizing the hero locations of 50 different trajectories. These trajectories were generated using the *Level-5 Model* with identical lineups on both sides, consisting of *zhaoyun*, *diaochan*, and *liyuanfang*, representing the *Assassins*, *Mages*, and *Marksmen* respectively. As depicted in Fig. 11, the heatmap is based on the HoK3v3 map.

When examining the *Assassins*, exemplified by *zhaoyun* (Fig. 11(a)), the heatmap illustrates the trajectory of *zhaoyun* as it permeates the entire map. This aligns with the role of *Assassins*, who are tasked with eliminating monsters throughout the jungle, acquiring gold and experience, and collaborating with *Mages* and *Marksmen* to eliminate opposing heroes. In the case of *Mages*, represented by *diaochan* (Fig. 11(b)), the heatmap reveals that *diaochan* predominantly remains within the middle lane, occasionally venturing into the lower section of the jungle to accrue resources by slaying monsters. As for *Marksmen*, embodied by *liyuanfang* (Fig. 11(c)), the heatmap demonstrates that *liyuanfang* primarily operates within the middle lane, but occasionally ventures into the upper region of the jungle to obtain gold and experience by dispatching monsters.

These findings indicate that *Mages*, such as *diaochan*, and *Marksmen*, like *liyuanfang*, strategically exploit resources in different areas of the jungle, specifically the lower and upper sections respectively, to mitigate competition for jungle resources. In summary, the map is effectively utilized by all three hero roles, each exhibiting distinctive characteristics.

### H.2    Ablation Study on Whether to Use Shared Rewards

For the experiments conducted in our paper, we utilized the **PPO** and **CPPO** algorithms. In each lineup, consisting of three heroes from both sides, we employed separate rewards for training purposes. In other words, different heroes within each lineup received distinct rewards during training. Consequently, we conducted an ablation study using the **CPPO** algorithm to investigate the impact of utilizing averaged shared rewards during the training stage. The results are presented in Fig. 12(a),

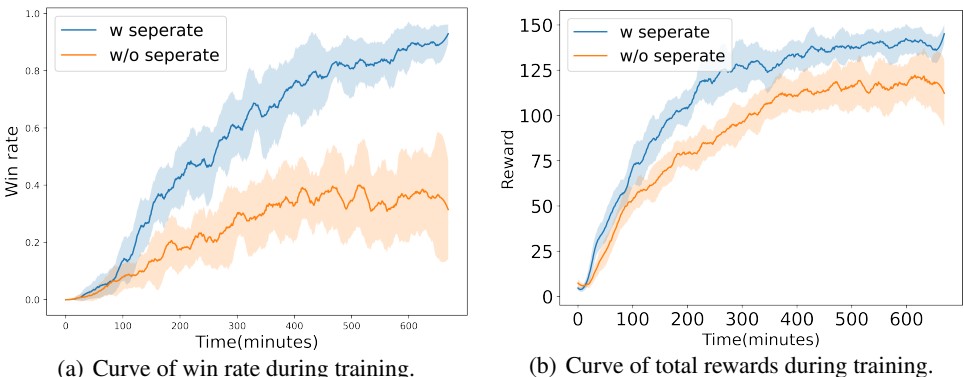

(a) Curve of win rate during training.    (b) Curve of total rewards during training.

Figure 12: Ablation study on whether to use shared rewards during the training stage under three random seeds. **w seperate**: heroes in every lineup using different rewards. **w/o seperate**: heroes in every lineup using the same averaged rewards.

which depicts the winning rate curve during training with both shared and no_shared rewards. The figure clearly illustrates a significant performance drop when using shared rewards, which can be attributed to the diverse roles of heroes, such as "*Marksmen*" and "*Mages*", who possess distinct fighting styles. Therefore, it is appropriate to assign separate rewards to each hero based on their specific roles. Additionally, as shown in Fig. 12(b), the total rewards obtained by heroes when using shared rewards are substantially lower compared to those using no_shared rewards. Consequently, for all other experiments, we employ no_shared rewards.

### H.3 Ablation Study on Whether to Use Zero-Sum Rewards

As described in Appendix. C.5, all the experiments conducted in our paper utilize zero-sum rewards to train, see Equation. 1. Therefore, we have conducted an ablation study to investigate the impact of using zero-sum rewards, as depicted in Fig.13. In other words, when employing zero-sum rewards, calculations are based on Equation. 1, whereas without zero-sum rewards, Equation. 6 is used. Fig. 13(a) demonstrates that during the later stages of training, the *w/o zero-sum* approach performs significantly worse than the **w zero-sum** approach, indicating that the utilization of zero-sum rewards can enhance the upper limit of the algorithm. Additionally, Fig. 13(b) illustrates that the total rewards obtained with zero-sum calculations are fewer compared to those obtained without zero-sum. This discrepancy arises from the subtraction of the average reward of the enemy camp (Equation. 1) when zero-sum rewards are employed. Consequently, for all the other experiments, we employ zero-sum rewards.

$$hero_{reward} = w_1 r_1 + w_2 r_2 + ... + w_n r_n \tag{6}$$

### H.4 Ablation Study on Whether to Use Sub-Action Mask

As described in Appendix C.4, sub-action mask is utilized to exclude certain *Action Types* that cannot be executed simultaneously with the current action. In other words, during each time step, not all *Action Types* (Table. 8), namely "Button", "Move", "Skill-X", "Skill-Z" and "Target" are necessary for training. Therefore, a sub-action mask is employed to eliminate the irrelevant *Action Types*. Therefore, an ablation study is conducted, as illustrated in Fig. 14, to examine the use of the sub-action mask. In Fig. 14(a), the winning rate of the *w/o SAM* (without sub-action mask) condition is considerably lower than that of the *w SAM* (with sub-action mask) condition. This discrepancy can be attributed to the fact that, during the backward stage of *w/o SAM*, gradients of the irrelevant *Action Types* can be regarded as noise interfering with the learning process of others. In Fig. 14(b), it can be observed

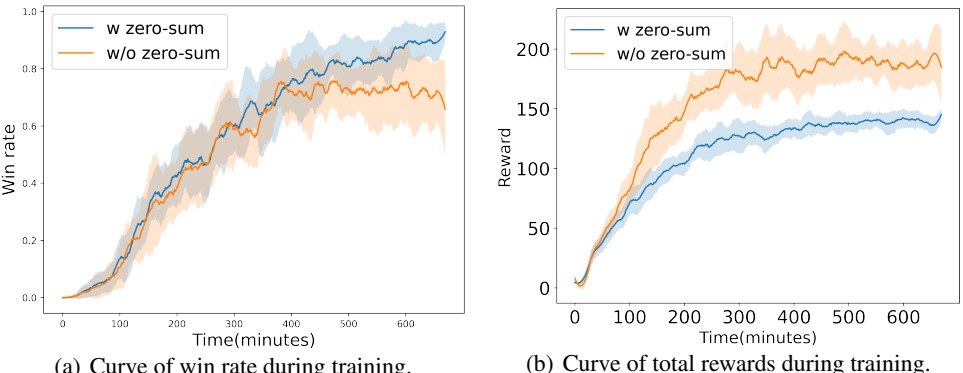

(a) Curve of win rate during training.  (b) Curve of total rewards during training.

Figure 13: Ablation study on whether to use zero-sum rewards during the training stage under three random seeds. **w zero-sum**: Rewards are caculated by Equation. 1. **w/o zero-sum**: Rewards are caculated by Equation. 6.

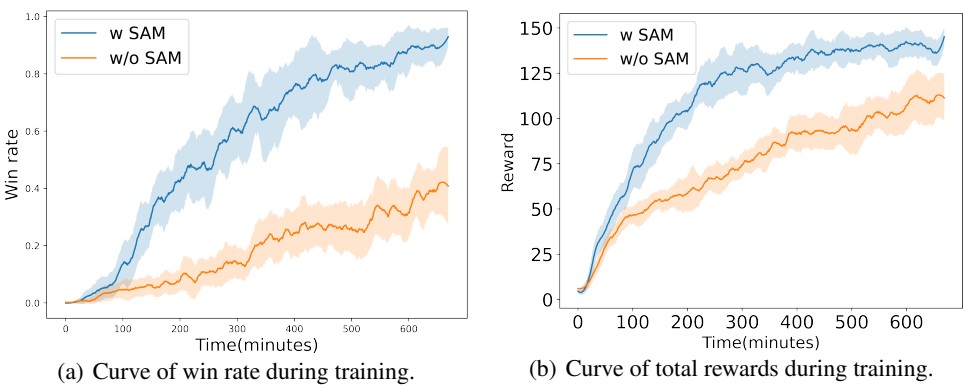

(a) Curve of win rate during training.  (b) Curve of total rewards during training.

Figure 14: Ablation study on whether to use sub-action mask(SAM) during the training stage under three random seeds. **w SAM**: Using sub-action mask during training. **w/o SAM**: Without sub-action mask during training.

that the total rewards obtained in the *w/o SAM* condition are fewer than those achieved in the *w SAM* condition, as expected. Consequently, for all the other experiments, we employ the sub-action mask.

## I   Limitations and Future Works

The complexity and realism of Honor of Kings provide more opportunities for diverse research directions, which have not been explored thoroughly and are crucial for our future work. We encourage broader community involvement in studying this environment. Besides, We plan to optimize the deployment of our environment to fit for multiple platforms and organize more competitions based on the Honor of Kings to expand the influence of the Honor of Kings environment and encourage research on reinforcement learning.

## J   Additional Discussion

### J.1   Discussion on the meaning os sub-tasks

Each individual subtask can be conceptualized as a sub-goal, representing a breakdown of the overall objective. For instance, the objective of destroying the enemies' crystal can be roughly broken down into sub-goals such as acquiring gold and experience points by defeating monsters -> hurt and kill

the enemies -> destroying their turrets -> ultimately shattering their crystal. This decomposition of the main goal has the potential to significantly enhance hierarchical and goal-driven research in multi-agent reinforcement learning (MARL). Moreover, these sub-goals possess semantic characteristics that allow them to be effectively linked with LLM agents.