# OpenReview forum: "HoK3v3: an Environment for Generalization in Heterogeneous Multi-agent Reinforcement Learning"
_NeurIPS.cc/2023/Track/Datasets_and_Benchmarks — Submitted to NeurIPS 2023 Datasets and Benchmarks_

### Official Review · Reviewer_naRj · 2023-06-25

**Rating:** 7
**Confidence:** 4

**Strengths:**

# significance

Good. The 3v3 environment incrementally extends the existing 1v1 environment. The key improvement is the combinatorial challenges of team composition. Previous environments have offered agent differentiation, but the magnitude of difference here is substantial.

Further, the GPU benchmarking shows the training is accessible to most researchers without large infrastructure investments. Typically complex game environments require compute resources outside the reach of small research groups or individuals, making this a valuable contribution in the space.

# relevance

Modest. The environment is for MARL, which is a sub-topic of RL research. The RL community has grown rapidly and MARL is a trending topic, so there is relevance to that group.

Arguably the greater relevance is supporting a "real world" environment. Relatively few research benchmarks (except perhaps SMAC or GRF) work with the full complexity of digital games that people play and this will enable work on far more complex domains with greater ecological validity. The lineup generalization results showcase that enabling agents to coordinate or compete with diverse teams is an open problem. The HoK 3v3 benchmark broadens the potential audience to many people in the games industry who may want to develop MARL algorithms on related games before they are able to work on their own (for example, during development).

# quality

Good. The experiments demonstrate that generalizing from competing against one lineup is not sufficient to compete against different lineups, and that controlling one lineup is not sufficient to control arbitrary other lineups. They also show the tasks can be solved by an oracle pretrained on these compositions, meaning the failures are not spurious effects of an impossible task. The algorithms are simple, but sufficient to demonstrate the key point for a benchmark.


# ethical and social implications

Poor. The paper does not include any discussion of the ethical implications in enabling agents to play human games, particularly given these agents could be readily used to replace people in the game. These can be both for good and ill, but merit some discussion given the game is structured to be played by 3 people vs 3 people.

**Additional Feedback:**

Questions:
- Are there GPU benchmark data available for other GPUs?
	- How easily can users configure the compute setup?
- Clarify that the six sub-tasks in figure 4 are trained from scratch each.
	- Or correct me if I am mistaken! It was not obvious from the text.
- Figures 5 & 6
	- In addition to numbering by "L-N", please indicate the number of heroes substituted. This would help interpret how performance degrades without trying to jump back and forth between the lineup definitions and the figures.
		- Consider subdividing the plot into sections for each number of substitutions.
	- "generalize to Marskmen heroes" - Which lineups are these?
		- Are these the "Archers" mentioned in the documentation?
			https://doc.aiarena.tencent.com/paper/hok3v3/latest/hok3v3_env/honor-of-kings/#hero-types
- What are the license requirements / access requirements for users?

**Clarity:**

Yes. The questions under "Additional Feedback" address some minor points of clarification.

**Correctness:**

Mostly yes. The evaluation methods use a rule-based baseline and reasonable PPO baseline to demonstrate training works. Generalization experiments follow the standard protocol of varying compositions.

The paper states differences among PPO, CPPO, and MAPPO, while the plots show overlapping intervals suggesting they are not significantly different. The means all converge to near 100% win rates.

**Documentation:**

The documentation website is thorough and provides additional information needed to understand the 3v3 environment differences (factorized reward and so on) from the 1v1 environment.

**Ethics:**

No.

The only question is what the criteria are for access to the environment.

**Limitations:**

Some prompting questions to consider on societal impact:
- What are the implications of agents replacing people in a multi-player game?
	- Should players know when this happens?
	- Should they be able to choose whether and how it occurs?
- What might be the effects of releasing these agents into the wild on how players learn and improve?
	- Could this change how players build skills in games?
	- Could it harm accessibility by making the barrier to entry too high?
- Do any of these concerns map to problems in other domains?
	- For example, human-robot interaction and teams.


**Opportunities For Improvement:**

# significance

Perhaps explain how the sub-task results are important. Why should the score when only considering one component of the reward be helpful? If the intent is to show this works as a curriculum, then it would help to show how this leads to positive transfer to the main task (ideally speedups that make up for the extra training time). Otherwise it is not clear how the sub-tasks are a significant contribution.

# relevance

The paper would benefit from connecting some of the generalization problems in (MA)RL to generalization problems in other research topics. Generalization is a broad topic of interest and the paper could highlight links to other efforts interested in the combinatorial discrete input setting (like open object vocabulary problems).

Additionally, the paper could link the notion of factorizable tasks to curriculum learning and other efforts to subdivide difficult tasks to enable better agent training and learning. For example, it seems the benchmark code does not need to change anything to support an automated reward reweighting.

# quality

Adding more benchmarking data for other GPUs or training on a typical workstation would help researchers understanding the scaling possible. Otherwise the quality is solid for the problem being addressed.

# ethical and social implications

As noted above, the paper would benefit from a discussion of the societal implications of replacing people with RL agents in a live game.

**Relation To Prior Work:**

Yes. The paper differentiates from other MARL environments and the previous 1v1 HoK environment.

**Summary And Contributions:**

The paper proposes a multi-agent reinforcement learning benchmark built on a widely played real world multiplayer game, Honor of Kings. The work extends a previous environment for 1v1 play in the game to support 3v3 competition, including play with teams of mixed character types. Compared to prior MARL work the introduction of characters with strongly distinct types adds complexity: both from differing capabilities of individual characters and their synergies on a 3-character team (particularly given the complex action and state space). Experiments demonstrate PPO, MAPPO, and a communication-based PPO variant are able to successfully train against a baseline AI agent. Tests show that there is a meaningful challenge to learning how to generalize to defeat different combinations of opponent teams and to control teams with varying mixes of characters. Training requires a single GPU along with a substantial pool of CPU cores (ideally over 256), making it accessible to a wide audience of researchers and practitioners.

---

> ### Author Response · Authors · 2023-08-23
> **Rebuttal**
>
> We would like to thank the review for the questions and suggestions. We provide a point-to-point response below.
>
> Q1: Perhaps explain how the sub-task results are important. Why should the score when only considering one component of the reward be helpful? If the intent is to show this works as a curriculum, then it would help to show how this leads to positive transfer to the main task (ideally speedups that make up for the extra training time). Otherwise it is not clear how the sub-tasks are a significant contribution.
>
> A1: The purpose of implementing sub-tasks is to ascertain the factorizability of our environment, HoK3v3, which holds significance in the field of Multi-Agent Reinforcement Learning (MARL) (see Section 2 of our paper). Specifically, the target of tasks in previous works, like SC2 or MPE, usually keeps integral and none of the current environments explicitly provide factorizable tasks, which lacks the support for hierarchical and goal-conditioned MARL. In addition, if the task is factorizable, it can better validate the performance of value decomposition, which is an important research area in MARL. In contrast, in one game of HoK3v3 we have a global target to destroy the crystal of the enemy, which can be naturally and explicitly factorized into several sub-tasks, including gaining gold, killing enemies, destroying the defense turrets, etc. Our implementation and experimental results also confirm the factorization. We acknowledge that curriculum learning may be a promising avenue for future research, but it is not the focus of our work.
>
> Q2: The paper would benefit from connecting some of the generalization problems in (MA)RL to generalization problems in other research topics. Generalization is a broad topic of interest and the paper could highlight links to other efforts interested in the combinatorial discrete input setting (like open object vocabulary problems).
>
> A2: Thank you for your constructive feedback. We recognize the importance of learning from generalization problems in other research domains and we are actively working on incorporating the generalization capabilities of large language models into (MA)RL. However, the primary objective of this paper is to introduce a new environment, HoK3v3, which, we believe, can offer suitable scenarios for assessing the efficacy of RL methods in addressing the issue of heterogeneous generalization. Therefore, we do not extensively discuss the connections between generalization problems in (MA)RL and those in other research areas in this work.
>
> Q3: The paper could link the notion of factorizable tasks to curriculum learning and other efforts to subdivide difficult tasks to enable better agent training and learning. For example, it seems the benchmark code does not need to change anything to support an automated reward reweighting.
>
> A3: Thank you for your valuable suggestion. As stated in Section 2 and the response to Q1, the implementation of sub-tasks serves the purpose of evaluating the factorizability of our environment, HoK3v3. This factorizability holds significance in the field of Multi-Agent Reinforcement Learning (MARL). However, it is worth mentioning that the curriculum is not the primary focus of our work. Furthermore, as you have observed, our benchmark code simplifies the process of reward reweighting, facilitating its integration with the curriculum, which is essential for our future endeavors in training higher-level agents.
>
> Q4: The paper would benefit from a discussion of the societal implications of replacing people with RL agents in a live game.
>
> A4: In fact, rule-based agents or RL-based agents have already been properly deployed as necessary assistants in certain gameplay scenarios. Players are well aware of their presence in the game, and these agents have been favorably received and effectively employed, resulting in a beneficial rather than detrimental impact. We will add these discusssions into our paper. Also note that, such a discussion and user cases have also been discussed in previous research (see the Broader Impact Section of the NeurIPS paper titled "Towards playing full MOBA games with deep reinforcement learning"), we will also mention this in our paper.
>
> Q5: What are the criteria for accessing to the environment？
>
> A5: There are no additional conditions required. Please refer to our comprehensive guidelines for easy utilization. Should you have any questions in following our documentation, please let us know. We will response soonest.
>
> Q6: Are there GPU benchmark data available for other GPUs? How easily can users configure the compute setup?
>
> A6: Currently there are no GPU benchmark for other GPUs, but users can easily configure the setup with no cost. Please refer to our step-by-step instructions provided in our documentation. As necessary, we can run experiments by varying the number GPUs used (we would like to follow your suggestions regarding this).
>
> ***TO BE CONTINUED***

---

> ### Author Response · Authors · 2023-08-23
> **Rebuttal**
>
> Q7: Clarify that the six sub-tasks in figure 4 are trained from scratch each.
>
> A7: Yes, they are trained from scratch.
>
> Q8: Figures 5 & 6，In addition to numbering by "L-N", please indicate the number of heroes substituted. This would help interpret how performance degrades without trying to jump back and forth between the lineup definitions and the figures. Consider subdividing the plot into sections for each number of substitutions.
>
> A8: Thank you for helping improve our paper. We will revise accordingly.
>
> Q9: "generalize to Marskmen heroes" - Which lineups are these?Are these the "Archers" mentioned in the documentation?
>
> A9: Yes. Marksman is Archer. We use them interchangeably. Sorry for the confusion caused. We will keep the term consistent in the doc.
>
> Q10: What are the license requirements / access requirements for users?
>
> A10: Our HoK3v3 environment follows the Apache-2.0 license. The game engine is publicly downloadable at: https://aiarena.tencent.com
>
> ***END***

---

> > ### Comment · Reviewer_naRj · 2023-08-26
> >
> > Thank you for the thorough replies! I'll only address the questions that I wanted to explore a bit further:
> >
> > > A1: The purpose of implementing sub-tasks is to ascertain the factorizability of our environment, HoK3v3, which holds significance in the field of Multi-Agent Reinforcement Learning (MARL) (see Section 2 of our paper).
> >
> > I'm not sure I follow how defining scores based on gold gained or enemies killed is a factorization that is useful for studying MARL, compared to the overall objective of capturing the enemy tower. What scientific hypotheses can we explore using these sub-tasks that is not available through the main task? What new methods can we test that would otherwise be untestable?
> > Section 2 was not clear on these points, so I have not been able to understand why the form of sub-tasks here is valuable. Concrete examples will likely answer this question and would be good to include in the paper to make this more obvious to other readers.
> >
> > > A4: In fact, rule-based agents or RL-based agents have already been properly deployed as necessary assistants in certain gameplay scenarios. Players are well aware of their presence in the game, and these agents have been favorably received and effectively employed, resulting in a beneficial rather than detrimental impact.
> >
> > Great to hear! This would be good to highlight how they are socially beneficial and what measures were taken to avoid detrimental effects. Games are often the leaders in new social integrations with technology, so the experiences with HoK are valuable to highlight and share for others in the research community to study.
> >
> > > A2: ...the primary objective of this paper is to introduce a new environment, HoK3v3, which, we believe, can offer suitable scenarios for assessing the efficacy of RL methods in addressing the issue of heterogeneous generalization.
> >
> > I appreciate the focus on the environment. I still would encourage some discussion that connects the idea of heterogeneous generalization to notions of out-of-domain generalization or other forms of generalization as used in related literature. If the environment is intended to assess generalization, then related efforts to assess generalization are a core part of what the paper should reference.

---

> > > ### Author Response · Authors · 2023-08-29
> > >
> > > Thank you for your further feedback. We discuss the points you mentioned below.
> > >
> > > Q1: I'm not sure I follow how defining scores based on gold gained or enemies killed is a factorization that is useful for studying MARL, compared to the overall objective of capturing the enemy tower. What scientific hypotheses can we explore using these sub-tasks that is not available through the main task? What new methods can we test that would otherwise be untestable? Section 2 was not clear on these points, so I have not been able to understand why the form of sub-tasks here is valuable. Concrete examples will likely answer this question and would be good to include in the paper to make this more obvious to other readers.
> > >
> > > A1: Thank you so much for your suggestions to make our paper clearer. Each individual subtask can be conceptualized as a sub-goal, representing a breakdown of the overall objective. For instance, the objective of destroying the enemies' crystal can be roughly broken down into sub-goals such as acquiring gold and experience points by defeating monsters -> hurt and kill the enemies -> destroying their turrets -> ultimately shattering their crystal. Such a decomposition of the main goal has the potential to facilitate hierarchical and goal-driven research in multi-agent reinforcement learning (MARL). Moreover, these sub-goals possess semantic characteristics that allow them to be effectively linked with LLM agents. To make it clear in our paper, we have also added a further discussion on the meaning of these sub-tasks in Appendix J.
> > >
> > > Q2: This would be good to highlight how they are socially beneficial and what measures were taken to avoid detrimental effects. Games are often the leaders in new social integrations with technology, so the experiences with HoK are valuable to highlight and share for others in the research community to study.
> > >
> > > A2: In our gameplay scenario, learning based agents have been deployed to help improve the player experience, including but not limited to: 1) notifying the human teammates when a player disconnects whether to host the disconnected player (note that it is not uncommon to see players drop offline during a started game, a.k.a, AFK (away-from-keyboard), especially for mobile games). In such cases, agents have been well designed to be at the same level as the offline player, limiting the agent to maintain a close perception and response speed to humans, and ensuring that the agent ​does not destroy the game environment through various mechanisms and game balance. 2) During the game development process, we use learning-based agents to conduct game balance tests, guide the numerical design of new game characters, and ensure that new game characters will not destroy the game balance. 3) We have also practiced a new game mode, called the JueWu AI Challenge, where players are allowed to fight against a team of five agents, with 20 levels of difficulty, which has attracted tons of players to participate.
> > >
> > > Q3: I appreciate the focus on the environment. I still would encourage some discussion that connects the idea of heterogeneous generalization to notions of out-of-domain generalization or other forms of generalization as used in related literature. If the environment is intended to assess generalization, then related efforts to assess generalization are a core part of what the paper should reference.
> > >
> > > A3: Agree. We intend to add one more pagragraph in the related work section to discuss the linkage of heterogeneous generalization and other categories of generalization in RL, such as the type of generalization, i.e., model generalization across multiple heroes, studied in the previously published HoK1v1 paper, and papers that measure generalization issues in RL. In case that we miss some important references, we would like to sincerely seek for your further suggestions on: which environments or technical papers that address RL generalization you would recommend us to add into our references?

---

### Official Review · Reviewer_y5Qe · 2023-07-07
**review for HoK3v3: an Environment for Generalization in Heterogeneous Multi-agent Reinforcement Learning**

**Rating:** 6
**Confidence:** 4
**Clarity:** The paper writing is clear.

**Strengths:**

- The environment is relatively complex in terms of agent type and actions space.

- The environment can be an alternative for The StarCraft Multi-Agent and Dota2

**Additional Feedback:**

no

**Correctness:**

The authors introduce a lot of subtask rewards, however, no information indicates that the sum of sub-task rewards monotonically increases with the winning rate. This sub-task reward may lead to a local optimal point.

The authors designed different sets up of lines up during training, which does not good for generalization. The generalization problem proposed by the authors may be located in the training process. It will be better to select random lines up in each or multiple episodes during the training.

**Documentation:**

The documentation for this infrastructure is good for Windows, but limited for Linux.

**Ethics:**

No.

**Limitations:**

The comparison of the proposed environment and related environment in terms of vectorization capability, speed, action space size, and agent types are limited.




**Opportunities For Improvement:**

- The state-of-art of algorithms can solve multiplayer games such as 5v5 dota2 and starcraft. Therefore the claim on limitation of “e limitations of existing RL methods” is overclaimed.
- The reviewer finds it hard to understand what is hierarchical action space. Is it simply a vector of 4 continuous actions [5, 4, 19,3]?
- Why the authors do not use Value-Decomposition Networks RL.
- The authors could list a table indicating a detailed comparison of the proposed environment and other related works.

**Relation To Prior Work:**

It is not clear how this work is better than the environment in Dota 2 with Large Scale Deep Reinforcement Learning

It is better to include the comparison of the proposed environment and related environment in terms of vectorization capability, speed, action space size, and agent types are limited.

**Summary And Contributions:**

This paper introduces a heterogeneous multiple-agent environment based on the game of honor of kings.

Two main contributions of the environment are:
- The scenario is relatively complex but limited to 3v3 instead of 5v5.
- There are 1000 different lineups to evaluate the generalization capability.

There are limited comparisons of the speed of the environment and the vectorization capability of the environment, which are one of the main obstacles to such kind of environment.

---

> ### Author Response · Authors · 2023-08-23
> **Rebuttal**
>
> We would like to thank the review for the questions and suggestions. We provide a point-to-point response below.
>
> Q1: The state-of-art of algorithms can solve multiplayer games such as 5v5 dota2 and starcraft. Therefore the claim on limitation of “e limitations of existing RL methods” is overclaimed.
> A1：Thanks for the suggestion. We also agree that our current claim regarding "limitations of existing RL methods" is not proper. We have fixed it in the updated version.
>
> Q2: The reviewer finds it hard to understand what is hierarchical action space. Is it simply a vector of 4 continuous actions [5, 4, 19,3]?
>
> A2: Sorry for the confusion caused. Please refer to Section C.2 in the Appendix, where we have provided a detailed explantion to what hierarchical action space refers to. Here we would like to provide an illustration in the rebuttal: think of a hero that has two actions, say move and fight. For move, the hero can move to 4 directions, let's say up/down/left/right. Fro fight, the hero needs to choose who to fight against in a certain game scene.  Adopting our hierarchial action space, in this case, we have two hierarchies, where the first hierarchy consists of two actions, i.e., move and fight, while the second hierarchy specifies how to execute a move a or a fight action.
>
> Q3: Why the authors do not use Value-Decomposition Networks RL.
>
> A3: Note that in this paper, we emphasize standardization and openness of a MOBA game environment to the community. We implement the most widely used algorithm in the literature of AI for MOBA game playing, i.e., PPO, as a benchmarking example. We leave it as a future work to dig out the potential of Value-decomposition networks. We also encourage the community to try out different algorithms using our environment.
>
> Q4: The authors could list a table indicating a detailed comparison of the proposed environment and other related works.
>
> A4: As per your suggestion, we have included a comprehensive table that compares HoK3v3 with other related works in terms of observation space, action space, focus and agent types in Appendix C.6.
>
> Q5: The comparison of the proposed environment and related environment in terms of vectorization capability, speed, action space size, and agent types are limited.
>
> A5: Thank you for your suggestion. Note that our MOBA game environment is the first open environment to the RL community. The most related environment is the DOTA 2 game environment used by OpenAI, which is not publicly available. The other similar environment includes Starcraft. However, StarCraft belongs to RTS games, which is of a different game genre. The stats of its game engine is not comparable to a MOBA game environment, e.g., vectorization capability, speed, etc. Furthermore, the concrete info of the DOTA 2 environment and the StarCraft engine are also not publicly available. We argue that while this paper aim at providing the community with a new MARL environment that is distilled from a complex, modern, and very popular MOBA game, Honor of Kings, it is out of focus to compare the action space, speed with other environments (we welcome further discussions on this point). For agent/hero types, the hero types mentioned in this paper are same as those of DOTA 2.
>
> Q6: The authors introduce a lot of subtask rewards, however, no information indicates that the sum of sub-task rewards monotonically increases with the winning rate. This sub-task reward may lead to a local optimal point.
>
> A6: Thank you for raising this question. As discussed in Section 2 of the paper, the purpose of implementing sub-tasks is to ascertain the factorizability of our environment, HoK3v3, which holds significance in the field of Multi-Agent Reinforcement Learning (MARL). Thus, the concern about "sub-task reward may lead to a local optimal point" is not the central focus of our work.
>
> Q7: The authors designed different sets up of lines up during training, which does not good for generalization. The generalization problem proposed by the authors may be located in the training process. It will be better to select random lines up in each or multiple episodes during the training.
>
> A7: The generalization in our work implies that policies trained on only one or a few lineups need to generalize on different heroes to adapt for various agent and opponent lineup during evaluation. Thus, in our experimental settings, we evaluate the performance of two trained policies during evaluation: the zero-shot policy, which is trained on the default lineup, and the multi-task policy, which is trained on a few different lineups.
>
> ***TO BE CONTINUED***

---

> ### Author Response · Authors · 2023-08-23
> **Rebuttal**
>
> Q8: It is not clear how this work is better than the environment in Dota 2 with Large Scale Deep Reinforcement Learning.
>
> A8: We double checked from the DOTA 2 official website and the OpenAI's DOTA 2 paper, we confirm that this work is the first ever MOBA environment publicly available. Our work wraps up one of the most popular MOBA games, Honor of Kings, into an easy-to-run environment, for the community. We provide step-to-step tutorial to enable more researchers and practitioners to have a change to access complex games like Honor of Kings. As a direct comparison with the DOTA 2 environment is not possible (as stated in A5 above), we believe this work advances in the spirit of openness and ease-of-use.
>
> Q9: The documentation for this infrastructure is good for Windows, but limited for Linux.
>
> A9: The only distinction between deploying on Windows and Linux is the requirement to build an additional gamecore image on Linux. We have provided detailed guidance on this matter, which can be found at:  https://github.com/tencent-ailab/hok_env/blob/master/docs/run_windows_gamecore_on_linux.md
>
> ***END***

---

### Official Review · Reviewer_6TCe · 2023-07-20
**HoK3v3 Review**

**Rating:** 5
**Confidence:** 4
**Correctness:** The experiment design is appropriate …

**Strengths:**

Zero-shot generalization in MARL is an emerging area of research which might be accelerated with high-quality benchmarks. The proposed benchmark defines 4 zero-shot MARL generalization tasks composed of intuitively designed sub-tasks. Care is given to practical concerns of researchers providing a discussion of resource requirements and an associated guideline.

**Additional Feedback:**

The benchmark installation requires the installation and inter-operation of multiple operating systems. This might impact the adoption of the benchmark in favor of one with more straight-forward installation.

Is it possible to execute any of the current MARL libraries with the environment?

**Clarity:**

The writing was clear with few errors. Text in figures could be increased in size as it is difficult to read at present.

**Documentation:**

Documentation covers game mechanics and state/action/reward information. However, I could not find instructions which would reproduce the experiments presented in the paper. There are many configuration files present in the codebase without explanations. The pretrained models needed for evaluation are difficult to find, inspecting the commit altering the code repository for HoK 3v3 did not appear to include any files which might be the referred pretrained models.

**Ethics:**

I have no ethical concerns with this work.

**Limitations:**

The authors have addressed limitations in the appendix.

**Opportunities For Improvement:**

O1 Reproduction of the results of the experiments may offer significant difficulty.

O2 Clarity regarding contribution with respect to prior work of HoK 1v1 could be improved with further detail.

**Relation To Prior Work:**

It is unclear what delineates some of the new contributions compared to the prior work proposing HoK 1v1. In this work rewards are broken in 5 sub-task definitions with very similar definitions to those proposed here along with a hierarchical action space. A degree of the generalization challenges is also present in the 1v1 environment.

**Summary And Contributions:**

The proposed benchmark reinforcement learning environment extends the Honor of Kings 1v1 environment to 3v3. An API for MARL algorithms is provided together with a hierarchical action space and 6 sub-task definitions. A zero-shot generalization study is performed to demonstrate the difficulty of transferring learned policies from a subset of agent controlled heroes to a broader set. Experiments indicate generalization for 3 baseline methods is poor in the scenarios examined.

---

> ### Author Response · Authors · 2023-08-23
> **Rebuttal**
>
> We would like to thank the review for the questions and suggestions. We provide a point-to-point response below.
>
> Q1: Reproduction of the results of the experiments may offer significant difficulty.
>
> A1: We have provided open-source game environment in github together with the codes to reproduce ppo experimental results. The community should be easy to reproduce other results using different algorithms following our step-by-step tutorial in the README file. Once you have any difficulties in running experiments, please post an issue to our Git, and we will fix soonest. Note that this project and this environment will be maintained for long. Also note that we have already used the same environment to organize AI competitions for universities worldwide for 3 consecutive years. Accessibility and reproducibility are not issues.
>
> Q2:  Clarity regarding contribution with respect to prior work of HoK 1v1 could be improved with further detail.
>
> A2: Overall, HoK1v1 emphasizes on generalization in competitive reinforcement learning, while HoK3v3 focues on generalization in a team of heterogeneous agents. To expand, a) 3v3 environment is more difficult compared with 1v1, considering its larger observation and action space. PPO can easily achieve very good performance in 1v1 environment, so we would like to test its upper bound using a more challenging task, such as 3v3;  b) 3v3 requires cooperation between different agents; c) Besides the standard 3v3 task, we also provided split-up sub-tasks to diversify the environment. Please also refer to the comments of Reviewer naRj: "The 3v3 environment incrementally extends the existing 1v1 environment. The key improvement is the combinatorial challenges of team composition. Previous environments have offered agent differentiation, but the magnitude of difference here is substantial".
>
> Q3: Text in figures could be increased in size as it is difficult to read at present.
>
> A3: Thank you for helping improve our paper. We will revise accordingly.
>
> Q4: Documentation covers game mechanics and state/action/reward information. However, I could not find instructions which would reproduce the experiments presented in the paper. There are many configuration files present in the codebase without explanations. The pretrained models needed for evaluation are difficult to find, inspecting the commit altering the code repository for HoK 3v3 did not appear to include any files which might be the referred pretrained models.
>
> A4: Thank you for the suggestion. More detailed instructions will be provided in our document, including step-by-step tuturial on how to reproduce our experiments using PPO algorithm,and explanation of important configuration files. In addition, we will upload the pretrained models to our github repository.
>
> Q5: The benchmark installation requires the installation and inter-operation of multiple operating systems. This might impact the adoption of the benchmark in favor of one with more straight-forward installation. Is it possible to execute any of the current MARL libraries with the environment?
>
> A5: Good question. We are working on integrating current MARL libraries into the environment.

---

> > ### Comment · Reviewer_6TCe · 2023-08-31
> >
> > I understand that the results might be reproduced through further efforts. It remains unclear why a submission would not be wholly reproduced by the submitted source code.
> >
> > Unfortunately I am only able to review the material submitted at this time. As the pretrained models are part of the evaluation method suggested it seems these must be included. These models and additional documentation are not available for review at this time. Therefore, I maintain my original scoring.

---

> > > ### Author Response · Authors · 2023-08-31
> > >
> > > Thank you for your comments. We address your confusion below.
> > >
> > > Q1: As the pretrained models are part of the evaluation method suggested it seems these must be included. These models and additional documentation are not available for review at this time.
> > >
> > > A1: Agree. Following your advice, we have added these pretrained models and the corresponding usage into our Git. Please check it out: https://github.com/tencent-ailab/hok_env/releases. Now these models and additional docs are available for review.
> > >
> > > Q2: It remains unclear why a submission would not be wholly reproduced by the submitted source code.
> > >
> > > A2: We have made it wholly reproducible (see A1 and the newest Git: https://github.com/tencent-ailab/hok_env).

---

### Official Review · Reviewer_J3fJ · 2023-07-21
**This paper presents the introduction of HoK3v3, a 3v3 game environment designed specifically for conducting research on multi-agent reinforcement learning (MARL). The study evaluates the performance of three MARL algorithms on both the full task and sub-tasks. Additionally, the generalization capabilities of these three MARL algorithms are also examined.**

**Rating:** 6
**Confidence:** 4

**Strengths:**

1. The paper is well-written and easily understandable.
2. The task scenario of the environment is a typical MOBA game, which holds significance for research purposes.
3. The design and implementation of the environment appear to be comprehensive, and the paper provides a "python example" to facilitate easy utilization of the environment.
4. The paper provides crucial evaluation metrics for the environment, such as the calculation resources required, training hours, and sample frequency. These metrics are essential for assessing the environment's performance.


**Additional Feedback:**

The specific details can be referred to as "Opportunities For Improvement."

**Clarity:**

Well written and easy to follow.


**Correctness:**

The experimental section requires further supplementation, and the specific details can be referred to as "Opportunities For Improvement."


**Documentation:**

This paper provides the code for agent training and evaluation, and documentary via the links.


**Limitations:**

The paper does not explicitly discuss the limitations or potential negative societal impacts of their work. The research appears to have no apparent negative implications.


**Opportunities For Improvement:**

1. The paper only tests three MARL algorithms, namely PPO, CPPO, and MAPPO, which cannot fully represent the performance of all MARL algorithms. Could the authors provide additional baseline algorithms?
2. In Section 4, only PPO, CPPO, and MAPPO are compared with "common_ai." The performance of these three algorithms appears to be similar. Does this imply that most MARL algorithms can achieve similar levels of performance? What is the performance of "common_ai," and could it be considered a weak baseline? Could this experiment test a wider range of algorithm types to demonstrate the difficulty level of training in this environment?
3. In Section 5, "sub-tasks," I am unsure of the purpose of these experiments. If MARL algorithms already perform well on the full task (as shown in Figure 3), why is it necessary to train them on divided tasks? In Figure 4, CPPO outperforms other algorithms significantly in all sub-tasks, but its performance is similar to other algorithms in the full task, which raises questions for me.
4. In Section 6, "Generalization," why is the winning rate generally much higher in Varying Agent Lineups compared to Varying Opponent Lineups? Is it because Lineup 1 is too weak? Additionally, I believe that testing the generalization of only PPO, CPPO, and MAPPO in different Lineups may not adequately represent the performance of MARL algorithms. Why not include some algorithms specifically designed for task transfer problems in the testing?

**Relation To Prior Work:**

The previous work, HoK Arena, focused on the competitive setting within a 1v1 MOBA game, without taking into account heterogeneity in cooperation.


**Summary And Contributions:**

This paper introduces HoK3v3, a 3v3 game environment designed for conducting research in the field of multi-agent reinforcement learning (MARL). The paper provides a detailed description of the environment setup and its utilization. In terms of experiments, the study evaluates the performance of three MARL algorithms, namely PPO, CPPO, and MAPPO, on both the full task and sub-tasks. Furthermore, the study explores the generalization capabilities of these algorithms by conducting experiments in scenarios involving Varying Opponent Lineups and Varying Agent Lineups. While the writing style of the paper is clear and concise, the persuasiveness of the experiments could be enhanced by including more baseline algorithms for comparison.

---

> ### Author Response · Authors · 2023-08-23
> **Rebuttal**
>
> We would like to thank the review for the questions and suggestions. We provide a point-to-point response below.
>
> Q1: The paper only tests three MARL algorithms, namely PPO, CPPO, and MAPPO, which cannot fully represent the performance of all MARL algorithms. Could the authors provide additional baseline algorithms?
>
> A1：Yes we agree that they cannot fully represent all MARL algorithms. Considering the time limitation, we are unable to enumerate other MARL algorithms as additional baseline results. However, this will definitely be one of our future work, and hopefully the community can help us enrich the baseline results with the game environment and the source code we provided.
>
> Q2: In Section 4, only PPO, CPPO, and MAPPO are compared with "common_ai." The performance of these three algorithms appears to be similar. Does this imply that most MARL algorithms can achieve similar levels of performance? What is the performance of "common_ai," and could it be considered a weak baseline? Could this experiment test a wider range of algorithm types to demonstrate the difficulty level of training in this environment?
>
> A2: Common_ai is a rule-based agent, so its performance is relatively poor. The comparison between baseline results and common_ai is to prove the validity of algorithms. Hence, similar performance against common_ai does not imply similar ability of the tested 3 MARL algorithms. It is hard to explicitly demonstrate the difficulty level of our environment, which requires a lot of time and efforts. In the future, together with the efforts from the community, we believe more and more algorithms will be applied in our environment to generated diverse baseline results.
>
> Q3: In Section 5, "sub-tasks," I am unsure of the purpose of these experiments. If MARL algorithms already perform well on the full task (as shown in Figure 3), why is it necessary to train them on divided tasks? In Figure 4, CPPO outperforms other algorithms significantly in all sub-tasks, but its performance is similar to other algorithms in the full task, which raises questions for me.
>
> A3: The purpose of the two set of experiments are different. As mentioned, the experiments against common_ai is to prove the validity of algorithms, not to compare the performance among the 3 MARP algorithms. On the other hand, the divided sub-tasks have the following 2 purposes: 1) to verify the decomposability of our environment; 2) to provide easier and more diverse mini-tasks for researchers in their own algorithm development.
>
> Q4: In Section 6, "Generalization," why is the winning rate generally much higher in Varying Agent Lineups compared to Varying Opponent Lineups? Is it because Lineup 1 is too weak? Additionally, I believe that testing the generalization of only PPO, CPPO, and MAPPO in different Lineups may not adequately represent the performance of MARL algorithms. Why not include some algorithms specifically designed for task transfer problems in the testing?
>
> A4: Controlling different heroes in the same team to fight against the one fixed opponent, and controlling the same hero countering different opponents, can both test the generalization of the algorithm. For HoK3v3, the former is simpler than the latter, which is the reason why the winning rate  higher in Varying Agent Lineups compared to Varying Opponent Lineups. While this paper focuses on benchmarking representative MARL algorithms in our HoK3v3 environment, we leave it as a future work to include algorithms specifically designed for task transfer problems. We will add it into Conclusions and Future Work.

---

### Decision · Program_Chairs · 2023-09-22

**Decision:**

Reject

**Comment:**

This is a typical borderline paper:

On the one hand, reviewers liked that the paper is well written and presents an interesting benchmark that could be used for MARL research.

On the other hand, reviewers are concerned about the lack of baseline methods implemented in the benchmark and the fact that the environment cannot be integrated with other existing libraries. It also took the reviewers pointing out reproducibility challenges for the authors to provide the appropriate files and models. Lastly, there is a justified concern that the new benchmark is a rather small change compared to the 1v1 version.

Overall, I thus recommend rejection of the paper but could be swayed to acceptance.